# Cytotoxic Vδ2+ T cell subsets expand in response to malaria in human tonsil and spleen organoids

Kathleen D. Press[1], Florian Bach[1☉], Elsa Sola[2☉], Kylie Camanag[1], Nicholas L. Dooley[3,4], Anselma Ivanawati[3,4], Damian Oyong[3,4], Mayimuna Nalubega[3], Abel Kakuru[5], Sedrack Matsiko[5], Felistas Nankya[5], Kenneth Musinguzi[5], Annet Nalwoga[5], Evelyn Nansubuga[5], John Ategeka[5], Charles Ebusu[5], Bakar Odongo[5], Chloe Kashiwagi[1], Xuhuai Ji[6], Molly Miranda[6], Joselyn Tachiwa-Appiah[1], Kareena Sandhu[1], Lilit Kamalyan[2], Kattria van der Ploeg[1], Michelle Boyle[3,4], Lisa E. Wagar[7], Mark M. Davis[2]*, Prasanna Jagannathan[1]*

1 Department of Medicine, Stanford University, Stanford, California, United States of America, 2 Department of Microbiology and Immunology, Stanford University, Stanford, California, United States of America, 3 Cellular Responses to Diseases and Vaccination, Burnet Institute, Melbourne, Victoria, Australia, 4 Department of Immunology, School of Translational Medicine, Monash University, Melbourne, Victoria, Australia, 5 Infectious Diseases Research Collaboration, Kampala, Uganda, 6 The Human Immune Monitoring Center, Stanford University, Stanford, California, United States of America, 7 Department of Physiology & Biophysics, University of California Irvine, Irvine, California, United States of America

☉ These authors contributed equally to this work.
* mmdavis@stanford.edu (MMD); prasj@stanford.edu (PJ)

## Abstract

Vaccine effectiveness against malaria is dramatically reduced in malaria-exposed compared to malaria-naïve populations, potentially due to altered immune responses in secondary lymphoid organs following repeated infection. Newly developed human tonsil and spleen organoids, which replicate key features of B and T cell immunity, provide an exciting opportunity to overcome challenges of other models and to improve our understanding of innate-adaptive interactions in lymphoid tissue. The objectives of this study were to use these organoids to investigate the impact of malaria parasites on 1) cells within lymphoid tissues and 2) responses to a heterologous antigen. When we exposed organoids from malaria-naïve donors to *Plasmodium falciparum*-infected red blood cells (iRBC), we observed that iRBC exposure did not disrupt organoid formation and significantly increased Vδ2 + γδ T cell frequencies in both tonsil and spleen organoids at multiple timepoints. Single-cell RNA/TCR sequencing revealed that iRBC-responsive Vδ2 + T cells in organoids were clonally expanded and exhibited activated, cytotoxic phenotypes with upregulated expression of granzymes, interferon-stimulated genes, and antigen presentation machinery. TCR repertoire analysis demonstrated that malaria exposure drove clonal expansion of cytotoxic Vδ2 + T cells, contrasting with the diverse, smaller clones observed in control conditions. To validate these findings, we analyzed tonsils from Ugandan children

**Data availability statement:** All relevant data are within the manuscript and its Supporting information files. Single cell RNA/TCR sequencing data have been deposited in DRYAD and are available at DOI: 10.5061/dryad.3r2280gw3.

**Funding:** This work was supported by Stanford University, "Innovative Medicines Accelerator: Vaccine Prototyping using Tonsils-in-a-Dish", grant to KDP and PJ, by A.P. Giannini Fellowship to KDP, by Walter V. And Idun Berry Fellowship to FB, by CSL Centenary Fellowship to MB, by Snow Medical Foundation Fellowship, 2022/SF167, to MB. Additional support was provided for this work by Stanford Maternal and Child Health Research Institute, Woods Family Faculty Scholar in Pediatric Translational Medicine and by National Institutes of Health, R01 AI189963 to PJ. The Burnet Institute is supported by the NHMRC for Independent Research Institutes Infrastructure Support Scheme and the Victorian State Government Operational Infrastructure Support. The funders had no role in study design, data collection and analysis, decision to publish, or preparation of the manuscript.

**Competing interests:** The authors have declared that no competing interests exist.

with asymptomatic parasitemia and found expanded Vδ2+T cells with enhanced cytotoxic potential compared to uninfected controls. When we tested whether malaria pre-exposure affected subsequent recall responses to influenza vaccine, malaria pre-exposure or γδ T cell depletion did not significantly alter cellular frequencies or influenza-specific antibody responses in most donors, though modest reductions were observed in some individuals. This work demonstrates the utility of human lymphoid organoids for studying malaria-host interactions and provides novel insights into Vδ2+T cell biology, including evidence for clonal expansion and cytotoxic differentiation in response to malaria parasites within secondary lymphoid tissues.

## Author summary

*Plasmodium falciparum (Pf)* malaria vaccines are significantly less effective in populations with endemic malaria exposure compared to malaria-naïve individuals. We used human tonsil and spleen organoids to investigate whether *Pf* infection alters immune responses in secondary lymphoid organs, potentially contributing to this reduced vaccine efficacy. These organoids create a controlled system that preserves the architecture and cellular interactions of secondary lymphoid tissues. When we exposed organoids to *Pf*-infected red blood cells, we observed dramatic expansion of the Vδ2+subset of γδ T cells. This finding was particularly noteworthy because Vδ2+T cells are not typically considered major participants in immune responses within secondary lymphoid organs. Single-cell analysis revealed that these expanded Vδ2+T cells underwent clonal expansion and acquired cytotoxic phenotypes, suggesting antigen-specific responses. Tonsil tissue from Ugandan children with asymptomatic *Pf* infections showed similar patterns of Vδ2+T cell expansion and enhanced cytotoxic potential. Surprisingly, *Pf* pre-exposure did not affect subsequent recall responses to influenza vaccine in most donors, although this does not discount a possible impact on immune responses to primary vaccination. Our work reveals unexpected roles for γδ T cells in lymphoid tissues during *Pf* infection and establishes organoids as valuable models for studying host-pathogen interactions.

## Introduction

*Plasmodium falciparum* (*Pf)* malaria remains a major public health problem, with 600,000 deaths per year [1], predominantly among pregnant women and young children. Though two malaria vaccines, RTS,S and R21, are now approved by the World Health Organization, limited protective efficacy and rapid waning of protection highlight a need for more effective vaccines. A major challenge for malaria vaccine development has been drastic differences in efficacy between North American and African populations [2]. Though there are likely multiple factors contributing to reduced vaccine responses among African individuals, one hypothesis is that prior

or current infection with malaria may impair T and B cell interactions in secondary lymphoid organs, leading to impaired vaccine-induced immune responses [3,4].

The ability to generate protective B and T cell responses to infection and vaccination is critically dependent on innate immune responses. These innate immune cells support adaptive immunity by serving as antigen-presenting cells and producing cytokines that modulate germinal center reactions and promote antigen-specific antibody secretion. However, repeated malaria can significantly alter innate immunity (Reviewed in [3,4]). For example, semi-innate γδ T cells, particularly the Vδ2 + subset which dominate in circulation, proliferate following *in vitro* stimulation with *Plasmodium* parasites [5,6] or following acute symptomatic malaria [7,8], and kill parasites through a granulysin-dependent process [9]. Vδ2 + T cell frequencies correlate with protection from malaria both in natural infection [10,11] and in vaccination studies [12–15], but also contribute to inflammation, which has been implicated in driving more severe malarial disease [16,17]. Following repeated malaria, Vδ2 + T cell proliferation decreases [10,11,18,19] while Vδ1 + γδ T cells, which are more common in mucosal tissue and lymphoid organs, continue to expand and express more cytotoxic and activated phenotypes [18,20,21]. Indeed, the development of "clinical immunity" to malaria, or the ability to tolerate malaria parasites without developing symptoms, may rely in part in "tuning" these pro-inflammatory innate immune responses [10]. Whether these malaria-induced innate immune changes interfere with the development of protective B and T cell responses to infection and vaccination remains unclear. The lack of a model replicating immunity in secondary lymphoid organs have made answering this question particularly challenging to date.

Recently, human tonsil and spleen organoid systems have been developed which recapitulate key features of an adaptive immune response: production of antigen-specific antibodies, class-switch recombination, somatic hypermutation and affinity maturation as well as plasmablast differentiation. Critically these models replicate T-B cell interactions in the germinal center [22]. Tonsil and spleen organoids overcome some of the challenges of other systems; for example, animal models for malaria are limited in their ability to replicate chronic malaria and human studies with blood samples are limited in that circulating immune cells are distinct from immune cells in tissue or secondary lymphoid organs. To date, these organoids have largely been used to study secondary immune responses (i.e., responses to influenza antigen among people who previously received a flu vaccine) [22]. The organoid system provides a unique opportunity to characterize innate and adaptive immune responses following exposure to malaria parasites.

In this study, we investigated the impact of malaria parasites on human tonsil and spleen organoids. We observed that Vδ2 + γδ T cells proliferated in both spleen and tonsil organoids in response to *Pf*-infected red blood cells (iRBCs). Using single cell RNA/TCR sequencing, we identified that these cells were activated, cytotoxic and clonally expanded. Vδ2 + γδ T cells with cytotoxic phenotypes were also expanded within the tonsils of Ugandan children with a current *Pf* infection. When cells were exposed to both malaria parasites and flu vaccine (LAIV), malaria exposure did not significantly alter frequencies of B/T cell subsets or influenza-specific antibody responses. This work highlights a potential important function for clonally expanded, cytotoxic Vδ2 + γδ T cells in lymphoid organs following malaria exposure and supports further exploration into the impact of these cells on adaptive responses to malaria and other pathogens.

## Results

### Germinal centers form and Vδ2 + T cells expand following *Pf* exposure in tonsil and spleen organoids

Aiming to understand the impact of malaria on the lymphoid germinal center (GC) model, we exposed organoids to *Pf*-infected red blood cells (iRBC) or uninfected red blood cells (uRBC) and evaluated the impact on GC formation and resulting frequencies of immune cell subsets **(see** Fig 1A **for a schematic of the workflow.)** We utilized 13 tonsil samples (n = 9 age < 18, n = 4 age 18–30; 9 M, 4 F) and 3 spleens (male adults) from ongoing tissue collection cohorts at Stanford, and prepared organoids as previously described in a study assessing GC responses to influenza [22]. Importantly, tonsils enable studying responses in children (who are more severely impacted by malaria in endemic settings), while spleens have specific significance to malaria (malaria parasites sequester in spleens) and can be maintained longer in culture.

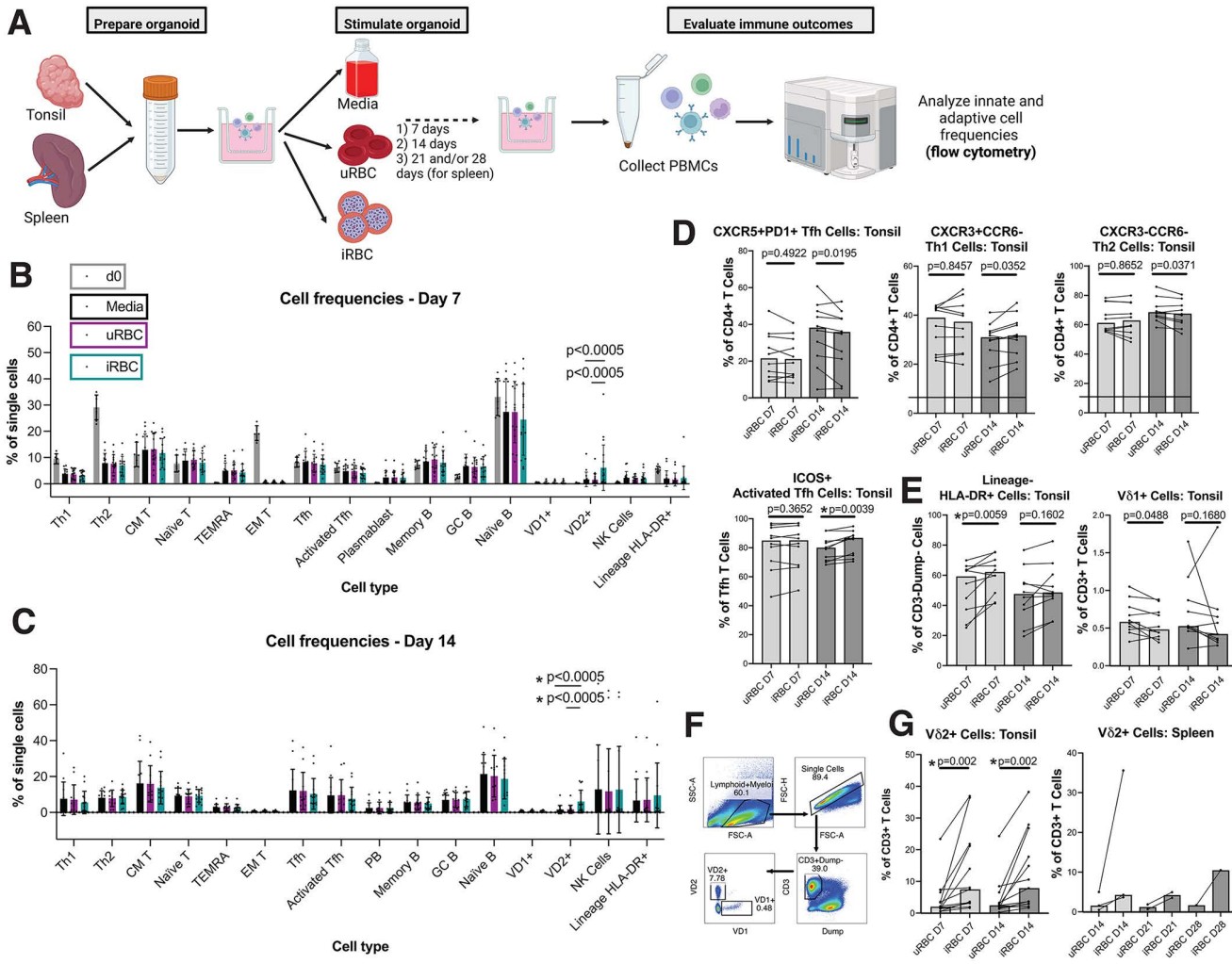

**Fig 1. Vδ2+T cell frequencies and phenotypes shift in response to iRBC in tonsil and spleen organoids. A.** Schematic showing experimental workflow for stimulation of organoid cultures. Created in BioRender. Dantzler, K. (2026) https://BioRender.com/t5k7t6a. **B-G**. Tonsil organoids (n = 10 for D7/D14; n = 7 for D0) were stimulated with uRBC or iRBC and cell frequencies were measured at day 7 and day 14. Cell frequencies were measured as a percentage of single cells (B-C) or of parent gate (D-F).-P-values were calculated using the Wilcoxon ranked sum test with correction for multiple comparisons using Benjamini-Hochberg method (FDR 0.05)Asterisks indicate p-values that remained significant after adjustment for multiple comparisons. **D-E**. Frequencies of some T cell subsets (D) and innate cells (E) shift in tonsil organoids following stimulation with iRBC compared to uRBC. **F.** Gating strategy for Vδ2+ and Vδ1+T cells. **G.** Vδ2+T cells proliferate following iRBC stimulation (G, n = 13). Bars show medians. **G.** In spleen organoids (n = 3), Vδ2+T cells are increased following iRBC (as compared to uRBC) at day 14, day 21 and day 28.

Cells reaggregated similarly in uRBC, iRBC, and untreated organoid conditions after 3 days (S1A Fig), indicating that GC formed successfully. Organoid cells were collected at day 7 and 14 for tonsil samples (which had dramatic cell death after day 14), and day 14 and 21 or 28 for spleens. To investigate the broad landscape of the resulting immune response, we compared total frequencies of T, B and innate cell subsets as a percentage of all singlet lymphoid/myeloid cells between media, uRBC-stimulated, and iRBC-stimulated conditions. We observed that the Vδ2+subset of γδ T cells were significantly higher in iRBC compared with uRBC or media conditions at day 7 (Fig 1B) and at day 14 (Fig 1C). Similar trends were observed for Vδ2+T cells in spleens collected at day 21 or 28 (S1B Fig). There were no significant differences

in frequencies of any other cell type between iRBC and control conditions (Fig 1B-1C). This increase in abundance of Vδ2+T cells in the iRBC condition strongly suggests that Vδ2+T cells respond specifically to malaria in the organoid system.

**Vδ2+T cells increase and change phenotype in response to iRBC in tonsil and spleen organoids**

In order to better understand changes driven by iRBC stimulation within specific T, B and innate cell subsets, we analyzed frequencies of cell subsets of their parent gate following uRBC and iRBC stimulation and compared the frequencies between the two stimulation conditions (see S1C-S1E Fig for gating strategies). Activated T follicular helper (Tfh) cell frequencies (as a proportion of Tfh) were higher for iRBC-stimulated compared to uRBC-stimulated cultures at day 14 (Fig 1D) and HLA-DR+ (a marker of activation and antigen presentation) innate cell frequencies (as a proportion of CD3-Dump- cells) were higher for iRBC-stimulated cultures at day 14 (Fig 1E). Additional cell frequencies (Tfh, Th1, Th2, Vδ1+T cells) trended towards differences between iRBC and uRBC stimulation at 1 timepoint but did not reach significance after correction for multiple comparisons (Fig 1D-1E). This observation is consistent with work showing that CD4+T cells expand and become activated in controlled human malaria infections (CHMI) [23,24]. Frequencies of memory T and B cell subsets and NK cells were not different between uRBC- and iRBC-stimulated conditions (S1F Fig). The only cell type that was significantly different between uRBC and iRBC stimulation at both timepoints was Vδ2+γδ T cells, which consistently increased in abundance in response to iRBC both in tonsils and spleens (Fig 1F-1G). This result is consistent with previous observations that in malaria-naïve individuals, Vδ2+T cell frequencies increase in response to both *in vitro* [5,6,25] and *in vivo* malaria exposure [7,8].

Given that different subsets of Vδ2+T cells may have diverse functions, we sought to characterize the phenotype of Vδ2+T cells induced by iRBCs. First, we examined expression HLA-DR on Vδ2+T cells from all tonsil donors, since HLA-DR was increased in Vδ2+T cells during *Pf* infection in Malawian children [19]. At both D7 and D14, Vδ2+T cells exposed to iRBC expressed significantly more HLA-DR compared to media controls; however differences between iRBC and uRBC were not significant (**S1G Fig**). Next, in a smaller experiment with 3 donors, we evaluated several known markers of T cell memory, activation and immune regulation and observed their expression on Vδ2+γδ T cells in the tonsil organoid culture. Tim-3, a marker of immune regulation, is involved in T cell exhaustion and may play a role in modulating Vδ2+T cells following recurrent malaria [10,11,26]. Tim-3 was higher in the iRBC-stimulated condition at day 7 compared to media/uRBC controls in 2/3 donors, and expression decreased from day 7–14 in all conditions (S1H Fig). CD8 expression was lower in the iRBC-stimulated condition at both day 7 and day 14 (**S1H Fig**). Central memory (CD45RA-CCR7+) populations decreased over time (S1H Fig) while effector memory (CD45RA-CCR7-) populations increased (S1H Fig) in all conditions, including uRBC. Together, these results suggest that iRBC exposure may influence Vδ2+T cell activation and phenotype in the organoid culture, including expression of HLA-DR, Tim-3, and CD8.

**Activated and cytotoxic Vδ2+T cell clusters expand in response to iRBC stimulation**

Aiming to more broadly characterize Vδ2+T cell phenotypes following iRBC stimulation, we stimulated spleen organoids from 3 donors for 10 days with iRBC or uRBC and performed single cell RNA/TCR sequencing on organoid cells, enriching for Vδ2+T cells (Fig 2A). Cells from day 0 served as a baseline control, along with peripheral blood mononuclear cells (PBMC) from 1 donor. Cells from all samples were then labeled with antibody-conjugated oligonucleotides, captured, and analyzed through the BD Rhapsody pipeline. We used antibody and RNA expression to define clusters of cell types that changed across the timepoints (Figs 2B and S2A-S2B). After quality control, the dataset comprised 64951 cells. We used Louvain clustering on 12 principal components to classify these cells into 22 populations and categorized populations using canonical cell type markers. When we summed frequencies of all clusters assigned to each cell type, we identified T cells (81.8%), B cells (11.6%), NK cells (3.8%) and a small population of erythrocytes. To disambiguate γδ from αβ T cell clusters, we adapted a classification approach [27] using gene modules comprised of TCR genes and both protein

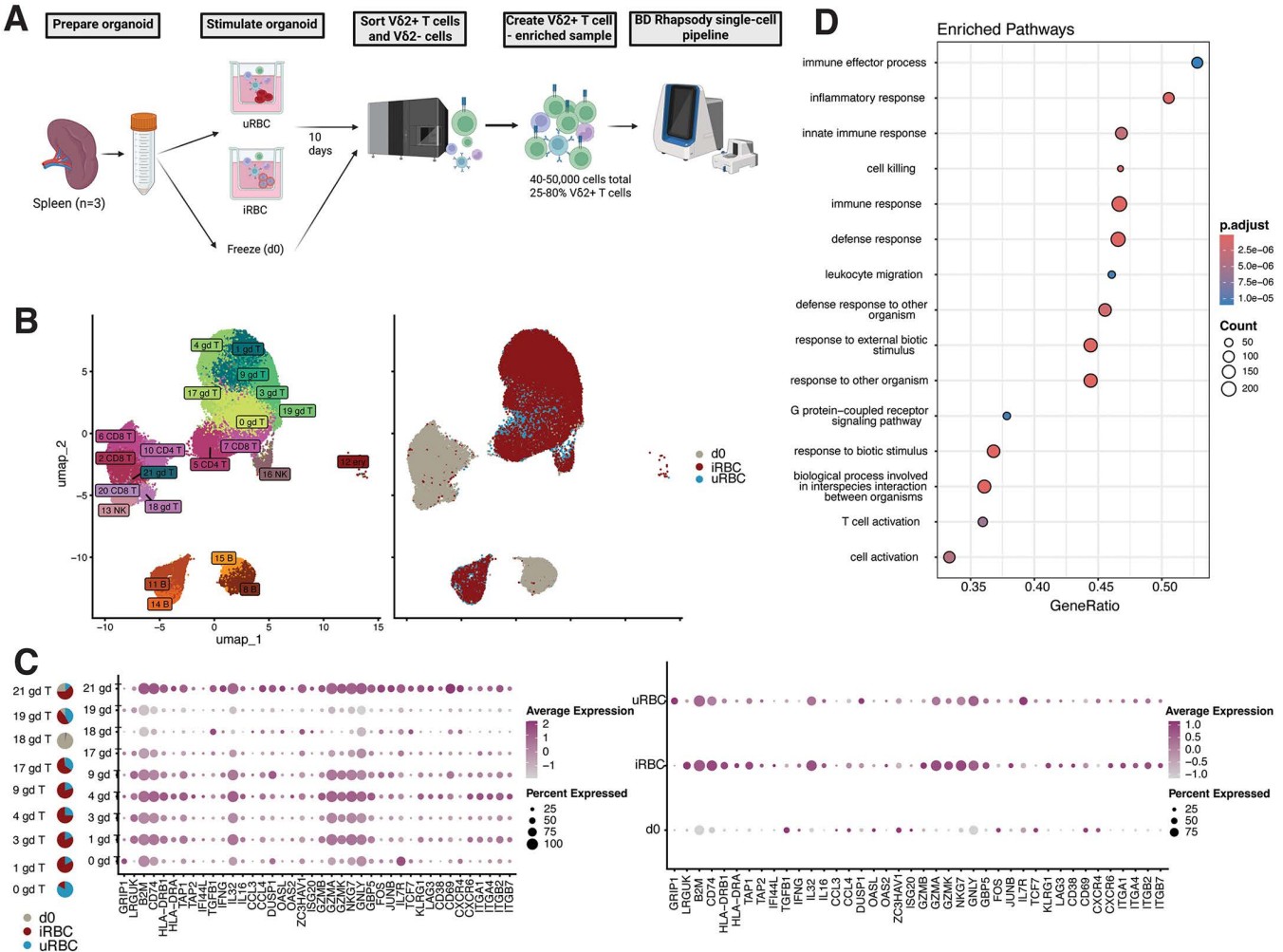

**Fig 2. Activated and cytotoxic Vδ2+T cell clusters expand in response to iRBC stimulation. A.** Schematic showing experimental design for iRBC stimulation of spleen organoids and Vδ2+ enrichment. For each sample, 8,000-40,000 Vδ2+ γδ T cells were sorted and depending on the number collected, 10,000-30,000 Vδ2- cells were sorted in order to reach a total of 40,000-50,000 cells per sample. Created in BioRender. Dantzler, **K.** (2026) https://BioRender.com/9jvreyv. **B.** UMAP projection of cells colored by Louvain cluster identity (left) or experimental condition (right). Specific genes that were used for annotating clusters are shown in S2A Fig. **C.** (Left) Heatmap showing select genes (x axis) that are specific to and variable between Vδ2+clusters (y axis). Pie charts show the relative distribution of cells from each experimental condition. (Right). Heatmap of the same genes shown across Vδ2+populations from each experimental condition. Dot size corresponds to percentage of cells in each cluster that express corresponding gene while color corresponds to intensity of expression. **D.** Top 15 most enriched GO terms based on signature genes from 3 γδ T cell clusters containing >75% cells from iRBC-stimulated condition (clusters 1, 3, and 9). Signature genes were defined separately for each individual cluster relative to all other clusters. Dot size corresponds to the number of signature genes (from clusters 1, 3, and 9) associated with each pathway. Color corresponds to level of significance that the list of signature genes is enriched for genes from the pathway.

and RNA expression of signature genes. This revealed 9 γδ clusters, two CD4 T and four CD8 T populations. Dominant γ and δ gene usage revealed that virtually all γδ T cells were Vδ2+γ9+ (S2C Fig). γδ T cell clusters 1–4 increased in abundance in response to iRBC and had an activated and cytotoxic phenotypes characteristic of effector T cells: loss of *TCF7* and *IL7R* expression, upregulation of activation markers (*CD38*, *HLA-DRB1*) and increased expression of terminal differentiation markers (*KLRG1*, *LAG3*) (Fig 2C). Furthermore, when comparing gene expression between Vδ2s from iRBC and uRBC stimulated conditions, expression of interferon-stimulated genes (*OASL*, *OAS2*, *IFI44L* and *GBP5),*

cytokines (*IFNG, IL16, IL32*) and cytotoxicity-associated molecules (*GZMA, GZMB, GZMK*) were increased in the iRBC stimulated conditions. Notably, markers of tissue residency, including a broad range of cell adhesion molecules (*ITGA1, ITGA4, ITGB2, ITGB7*) and *CXCR6* were upregulated in the iRBC-stimulated condition, as were a broad range of class I and class II antigen processing and presentation machinery (*B2M, CD74, TAP1, TAP2, HLA-DRB1, HLA-DRA HLA-DMA, HLA-DMB, HLA-DOB*) (Figs 2C and S2D). Additional markers of class II antigen presentation, including CIITA, HLA-DMA, and HLA-DMB were expressed slightly above background in γδ T cell cluster 1 (S2D Fig). This upregulation of antigen presentation genes is consistent with our observation that HLA-DR surface expression increased following iRBC stimulation (S1G Fig), suggesting that Vδ2 + T cells may not simply be responding to *P. falciparum* antigen, but may present antigen as well.

Consistent with iRBC-stimulated Vδ2 + T cells being highly activated and cytotoxic effector cells, gene ontology (GO) analysis showed enrichment of several GO terms associated with immune processes, including inflammation, innate immunity and cell killing (Fig 2D). Numerous genes associated with DNA replication and cell division were also upregulated among iRBC-responsive Vδ2 + T cells (S2E Fig). These results suggest that Vδ2 + T cells are the dominant T cell type responding to iRBC stimulation in lymphoid organoids. Their response is characterized by pro-inflammatory, cytotoxic and tissue residency gene expression.

## Vδ2 + T cells clonally expand in response to iRBC stimulation

Since we observed that Vδ2 + T cells increase in frequency in both tonsil and spleen organoids following iRBC stimulation (**Fig 1F**), we next investigated whether these cells were clonally expanded (adaptive-like), and thus possibly specific to iRBC antigens, or indiscriminately proliferated (innate-like). To address this, we analyzed the T cell receptor (TCR) repertoires of cells from iRBC compared to uRBC treated organoids. Clonotypes that were present both at day 10 of iRBC culture and at day 0 were significantly increased in response to iRBC stimulation, though the small size of the γδ T cell population at day 0 prevented full evaluation of TCR repertoire changes from day 0 to day 10 (Fig 3B). Smaller clones (0.01-0.1% and 0.1-1%) were more abundant in the uRBC condition (Fig 3A-3B), indicating higher diversity and less clonality. In contrast, iRBC stimulation consistently led to repertoires composed of larger clones (>1%) (Fig 3A-3B), suggesting higher clonal expansion and a more homogenous population of γδ T cells. Expanded clones corresponded to unique CDR3 sequences (S1 Table) and mapped to the cytotoxic γδ T cell clusters (Fig 3C). Conventional T cells after iRBC stimulation looked like γδ T cells after uRBC stimulation with a low relative contribution of larger clones and a more diverse repertoire of smaller clones (S3A Fig). The cumulative size of the largest conventional T cell clones did not vary much by stimulation condition (S3B Fig). Together, these results show that exposure to malaria antigens drives the clonal expansion of the cytotoxic Vδ2 + T cell repertoire in lymphoid organoids.

## Vδ2 + T cells with cytotoxic phenotypes are expanded in the lymphoid tissues in children with malaria

The organoid model represents an acute *Pf* infection while most individuals in malaria-endemic countries experience chronic, repeated infections. To assess if Vδ2 + T cells also expanded in lymphoid tissues during human malaria *in vivo*, we quantified Vδ2 + T cell frequency and activation in tonsils collected from children in a malaria endemic area of Uganda. All children were well at time of tonsillectomy, with asymptomatic parasitemia quantified by PCR of peripheral blood. Vδ2 + T cell phenotype and function were analyzed and compared between currently uninfected (n = 10, 5.5 [2.25-9.25] years (median [IQR]), 50% male) and asymptomatically infected children (n = 10, 6 [3.7-7] years (median [IQR]), 50% male) using flow cytometry (see **Fig 4A** for the gating strategy).

The Vδ2 + T cell (or Vδ2- T cell) proportion of total T cells was not increased in children with current asymptomatic parasitemia compared to uninfected (Figs 4B and S4A); however, Vδ2 + T cell proportions of T effector memory cells (TEM: CD27-CD45RA-) increased while naïve cells decreased (CD27 + CD45RA+) (**Fig 4C**), suggesting a shift towards a differentiated and cytotoxic phenotype. Intracellular production of Granzyme-B was increased in asymptomatic children

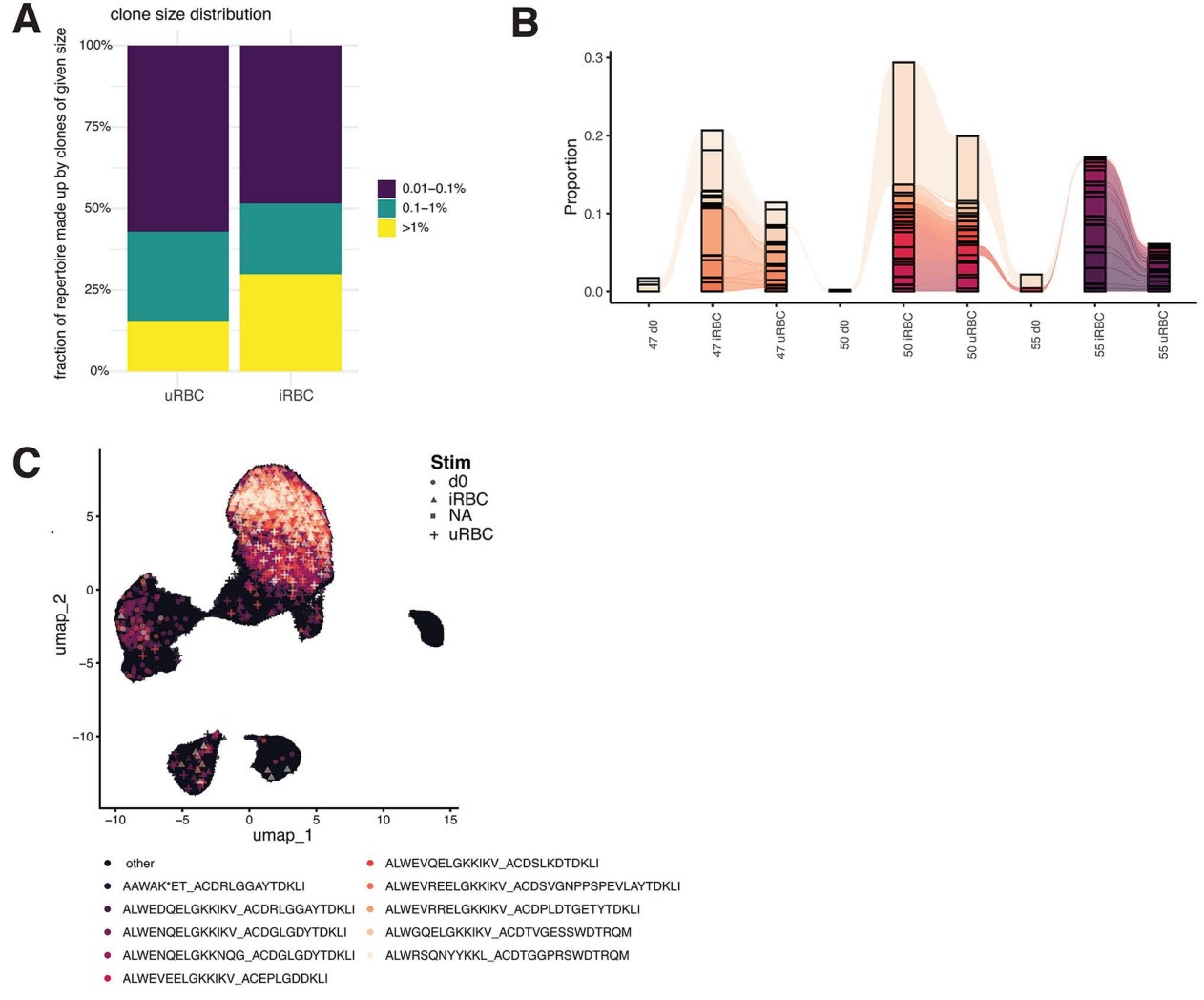

**Fig 3. Vδ2+T cells clonally expand in response to iRBC stimulation. A.** Bar graphs showing the percentage of each donor's Vδ2+T cell receptor repertoire occupied by clones of different orders of magnitude. A highly diverse repertoire corresponds to mostly small clones (dark blue) while a less diverse and highly clonal repertoire is made up of mostly large clones (yellow). **B.** Alluvial plot showing the distribution of the 50 largest Vδ2+T cell clones between donors (numbered 47, 50 and 55) and experimental conditions. Colors correspond to different clones, some of which are shared between donors. **C.** UMAP showing clone size corresponding to different stimulation conditions for the 10 largest Vδ2+TCR sequences.

compared to uninfected children, consistent with malaria-driven expansion of Vδ2+T cell with cytotoxic phenotypes *in vitro* (**Fig 4D**). To assess the inducible potential of tonsillar Vδ2+T cells, we quantified the overall cytotoxic activation following stimulation with PMA and ionomycin by Boolean gating. The magnitude of activation (CD38+), degranulation (CD107a+), Granzyme-B and pro-inflammatory cytokine (IFNγ and TNF) co-expression was increased in infected compared to uninfected children (**Fig 4E**). Frequencies of highly cytotoxic CD38+CD107a+Granzyme-B-IFNγ+TNF+ Vδ2+T cells were higher in infected compared to uninfected children (**Fig 4F**). These findings suggest that malaria increases the cytotoxicity of tonsillar Vδ2+T cells *in vivo*. Interestingly, frequencies of TEM, Granzyme B+, and CD38+CD107a+Granzyme-B+IFNγ+TNF+ subsets also increased in Vδ2- γδ T cells from infected children (S4B-S4E Fig), suggesting that in Ugandan tonsils, unlike in malaria-naïve organoids, cytotoxic non-Vδ2 γδ T cells expand following *Pf* infection, as well.

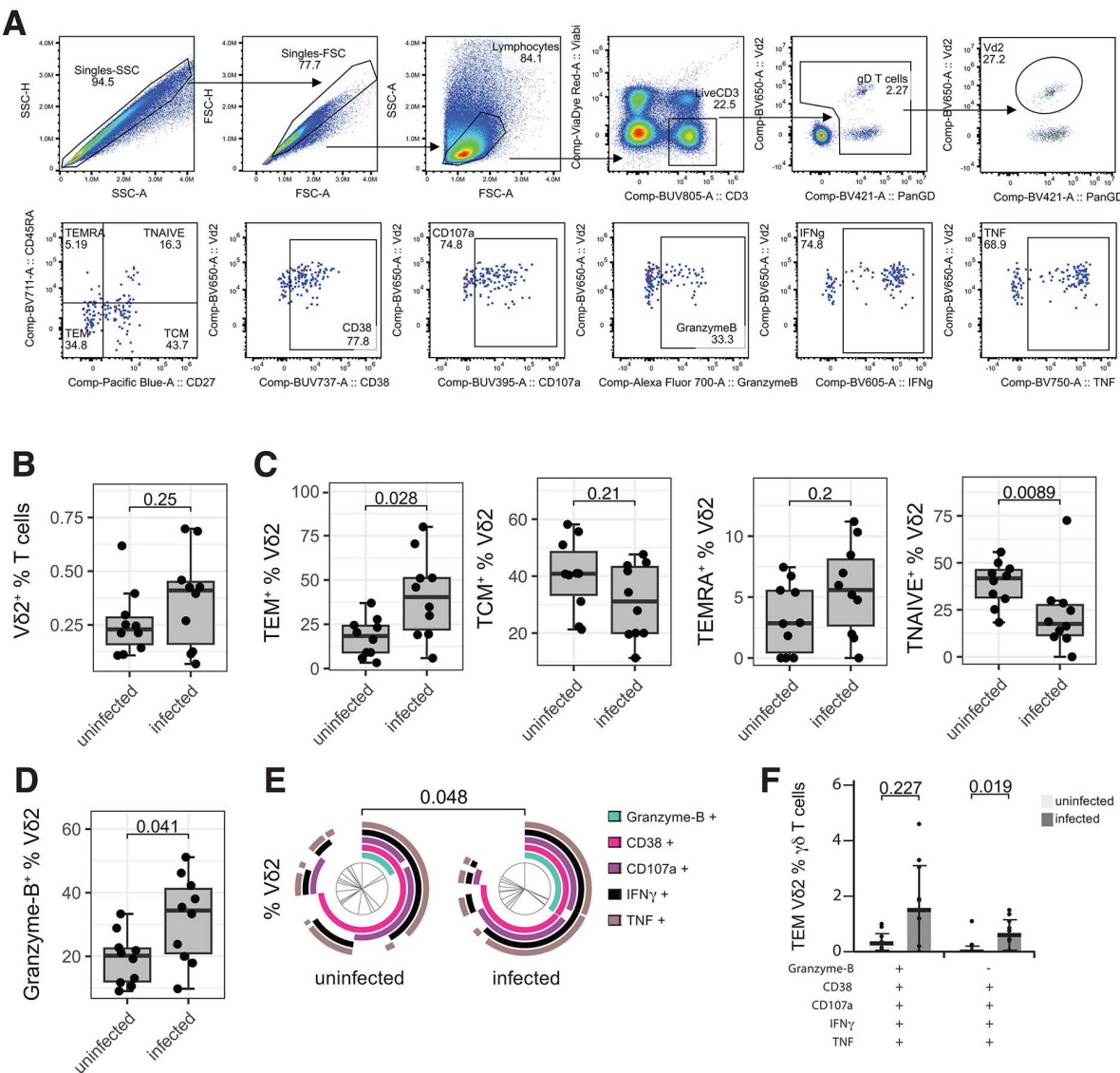

**Fig 4. Expansion of Vδ2+T cells with cytotoxic phenotypes in malaria infected children.** Isolated Tonsillar mononuclear cells from malaria-endemic Ugandan children with asymptomatic parasitemia (infected) or no *Pf* infection (uninfected) were stimulated with PMA and Ionomycin for 24 hours. **A.** Gating strategy for Vδ2+T cells and representative memory subset gating and marker expression. **B.** Vδ2+frequency of total T cells. **C.** Frequency of Vδ2+T effector memory (TEM: CD27-CD45RA-), T central memory (TCM: CD27+CD45RA-), T terminally differentiated effector memory (TEMRA: CD27-CD45RA+) and T *naive* (TNAIVE: CD27+CD45RA+). **D.** Frequency of Vδ2+T cell intracellular Granzyme-B expression. Comparisons performed by Mann-Whitney U test. Center line representing the median, box limits indicating the upper and lower quartiles, whiskers extending to 1.5 times the interquartile range. **E** Pie chart depicts co-expression of CD38, CD107a, Granzyme-B, IFNγ and TNF by Vδ2+T cells. Comparisons performed by permutation test. **F.** Frequency of CD38+CD107a+Granzyme-B+IFNγ+TNF+ and CD38+CD107a+Granzyme-B-IFNγ+TNF+ Vδ2+T cells. Bar representing mean and whiskers extend to standard deviation. Comparisons performed by Mann-Whitney U test.

## iRBC exposure does not impact cellular frequencies or influenza-specific antibody responses following LAIV vaccine exposure

Given established low vaccine responses in malaria-endemic populations, we hypothesized that exposure to malaria may impair memory T and B cell responses to other infections or vaccines. To address this question, we assessed cell

composition and specific antibody responses to an unrelated recall antigen, live-attenuated influenza vaccine (LAIV) in iRBC *vs.* uRBC treated organoids (see **Fig 5A** for schematic of the workflow). Immune cell frequencies were measured both as percentages of total single cells (S5A Fig) and percentages of parent gates (S5B-S5D Fig). We did not observe major changes in T cell, B cell or innate cell frequencies analyzed as percentage of parent gate between uRBC and iRBC conditions at either day 7 or day 14 (S5A-S5B Fig). Only Vδ2+T cell frequencies increased for iRBC+LAIV compared to uRBC+LAIV or LAIV alone (S5C Fig), which is due to the previously noted Vδ2+T cell expansion in all conditions containing iRBC (Figs 1F, S5A and S5B). Consistent with previously published data [22], we observed differential frequencies of numerous cell types (including memory T and B cell subsets, NK cells, and Vδ1+T cells) following LAIV stimulation compared to unstimulated controls. Vδ2+T cell frequencies did not change in response to LAIV when LAIV was added at d0 or d3 (S5D Fig), indicating that Vδ2+T cell expansion is specific to iRBC.

Next, we analyzed influenza-specific IgG antibody responses using Enzyme-Linked Immunosorbent Assay (ELISA) with supernatants from the stimulated organoid cultures (Fig 5A) and measured antibody responses as area under the curve

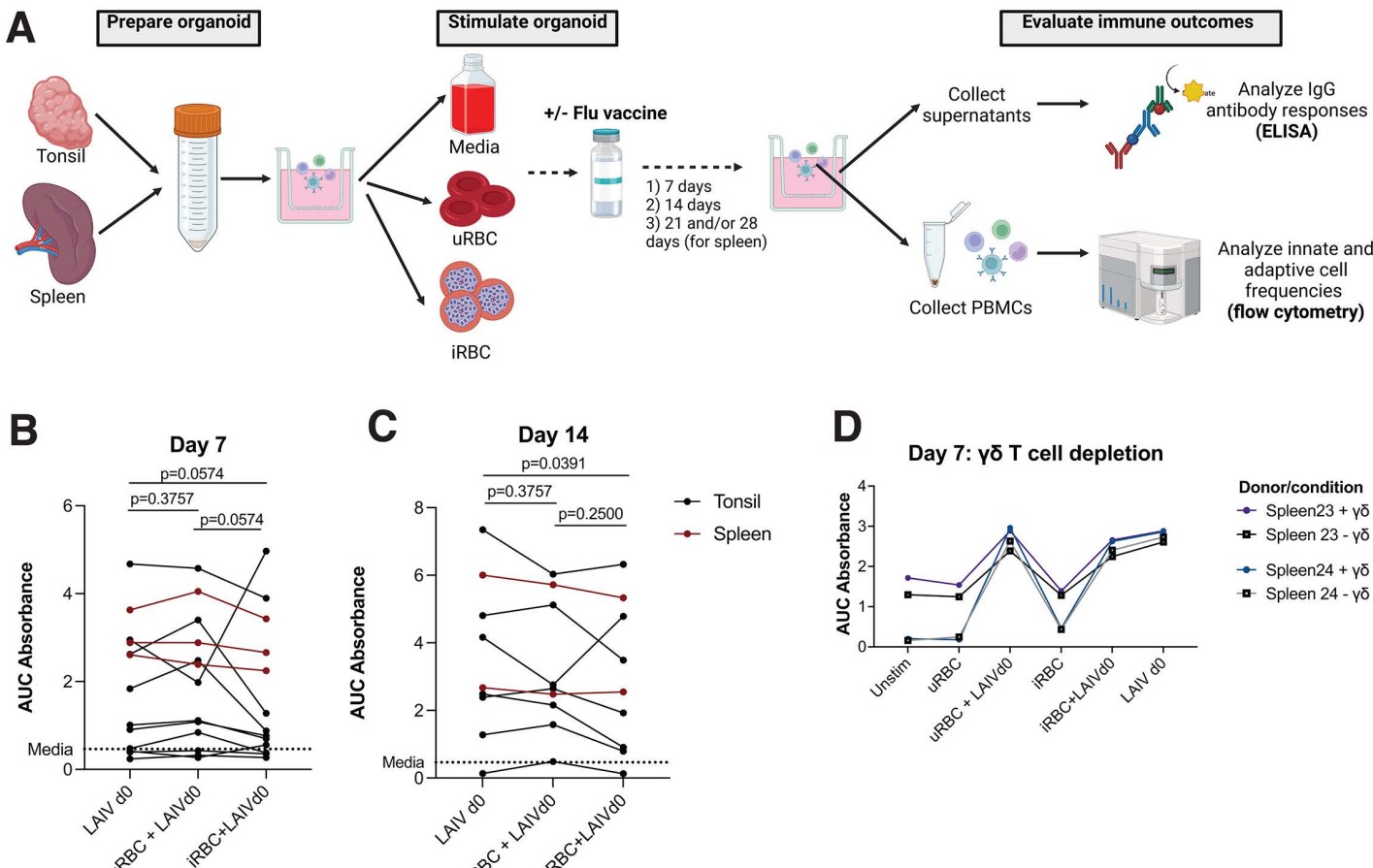

**Fig 5. Impact of iRBC exposure on flu antibody responses in tonsil and spleen organoids. A.** Schematic showing experiment workflow. Created in BioRender. Dantzler, K. (2026) https://BioRender.com/lgl72ow. Antibody responses to flu protein from ELISAs using supernatants from cultures stimulated with iRBC+LAIV compared to LAIV alone at Day 7 **(B)** or Day 14 **(C)**. Area under the curve (AUC) was calculated using the curve of absorbances across the 4 dilutions tested (neat, 1:10, 1:100, 1:300). AUC values above the average of media controls (0.463) were considered as detectable response. Tonsil donors are indicated in black and spleens in red. **D.** Antibody responses are no different between conditions containing γδ+T cells or depleted of γδ+T cells.

across the dilutions tested (AUC) [28]. At Day 7, iRBC + LAIV responses were not significantly different from uRBC + LAIV or LAIV alone (**Fig 5B**), while at Day 14, antibody responses were significantly lower in the iRBC + LAIV condition compared to LAIV alone (7 tonsils and 2 spleens) (P = 0.0391, **Figs 5C an**d S6A-S6B) but not compared to uRBC + LAIV (p = 0.2500). We did not observe any IgG antibody response (at any timepoint) to parasite schizont extract (S6C Fig) or specific parasite antigens (MSP1, MSP2, AMA1) (S6D Fig), but we did observe IgM responses to parasite schizont extract in half of the donors tested at day 7 (S6E Fig), and to a lesser extent on day 14 (S6F Fig). The organoid system could therefore be used to assess primary adaptive immune responses to *Plasmodium* parasites.

To determine whether the expansion of γδ T cells might alter influenza-specific antibody responses, we performed depletion experiments in 2 donors, where γδ T cells were antibody-depleted from spleen organoids prior to exposure to 7-day iRBC/uRBC stimulation. Although Vδ2 + T cells were significantly reduced (S7A Fig), antibody responses (Fig 5D), T and B cell frequencies (S7C-S7D Fig), and innate cell frequencies (S7E Fig) were similar between γδ T cell-depleted or non-depleted organoids. Together, these results do not support the hypothesis that exposure to malaria may impair responses to recall vaccination with influenza; however, this does not exclude the possibility of this relationship at other timepoints or following exposure to other pathogens/vaccines.

## Discussion

In this study, we found that Vδ2 + γδ T cells increase in abundance in tonsil and spleen organoids in response to malaria parasites, similarly to what has been observed in blood in malaria-naïve individuals. Despite a higher abundance of Vδ1 + T cells in tissues including spleen, Vδ1 + T cells did not expand in response to malaria parasites in tonsil and spleen organoids. γδ + T cells are unconventional T lymphocytes that uniquely bridge the innate and adaptive immune systems. They can play diverse roles, including production of pro-inflammatory cytokines, cytotoxic killing, antigen presentation, promotion of dendritic cell maturation, B cell help, recruitment of other immune cells, and secretion of growth factors [29]. γδ + T cells may also play a role in direct or indirect modulation of antibody production in secondary lymphoid organs [30–34]). They expand and kill parasites in response to natural *Pf* infection, CHMI, or vaccination with irradiated sporozoites, and are associated with protection in each context [10–15]. Current understanding of γδ + T cell subsets and phenotypes of subsets that proliferate and/or are associated with protection remains limited. The tonsil and spleen organoid systems provide a powerful way to analyze this important cell type in a fully human system.

Utilizing the organoid systems to functionally characterize Vδ2 + T cells in the context of a primary *Pf* infection, our results reveal a role for Vδ2 + T cells in secondary lymphoid organs—a novel finding given that Vδ2 + T cells are usually studied in the blood. These cells increased expression of HLA-DR, increased transcription of markers associated with activation and effector phenotypes, and were clonally expanded. Vδ2 + T cells in tonsils from Ugandan individuals with asymptomatic infection additionally became more differentiated and cytotoxic in response to secondary stimulation. An increase in expression of cytotoxic and activated markers on circulating Vδ2+ and Vδ2- T cells has been previously observed in active infections [19,35] while other studies report decreased pro-inflammatory markers and increased expression of immunoregulatory markers (TIM-3, CD57, CD16) among Vδ2 + T cells in children exposed to repeated malaria [10,11,26,36]. These data highlight the complexity of Vδ2 + T cell subsets, some of which may exert multiple functions or increase at different timepoints following an infection. Our results support diverse roles for Vδ2 + T cells in activation of T and B cells, and the observed upregulation of class I and class II antigen-processing genes in clusters enriched following iRBC stimulation warrants further investigation into Vδ2 + T cells as potential antigen-presenting cells. Mechanistic assays (using cells from both naïve and malaria-exposed individuals) could confirm that the observed Vδ2 + T cell activation is malaria specific and could further characterize the function of these cells including how this function may vary across donors, timepoints, or populations. It will be particularly interesting to evaluate how organoid Vδ2 + T cell phenotypes change in response to secondary stimulation with other pathogens or for longer stimulation periods. Additionally, as tools to image the organoids become more developed, it will be useful to examine the specific location of Vδ2 + T cells, as well as the cell-cell interactions involved in the function of this unique cell subset.

Our observation that Vδ2+T cells proliferating in response to malaria are clonally expanded in organoids provides a unique, important contribution to the field. The literature on γδ T cell clonal repertoires is very new, and so far has more consistently identified a larger role for clonal expansion among Vδ1+T cells (as compared to Vδ2+T cells), although one recent study did observe age-dependent clonal expansion in both subtypes [37]. In response to pathogens such as malaria or CMV, Vδ1+T cells differentiate [38–40] and transition to an effector T cell transcriptional profile [41]. By adulthood, the prevalent Vδ1+T cell receptor (TCR) repertoire becomes strongly focused on a few high-frequency clonotypes expressing cytotoxic effector T cell phenotypes [38]. Clonally expanded cytotoxic Vδ1+effector T cells are enriched in Malian children and adults [18] and in CHMI [42], they increase with repeated infections, undergo clonal selection, and differentiate into cytotoxic effector cells capable of responding to *Pf* parasites *in vitro* [18]. In contrast, Vδ2+T cells exhibited an initial robust polyclonal response to CHMI *Pf* infection that was not sustained with repeated infections [18]. We saw Vδ1+proliferation only in response to LAIV (particularly when added at day 3), and not in response to *Pf* parasites, which we speculate could be explained by the organoid representing a primary infection rather than a chronic infection. Our samples in the single-cell sequencing experiments did not contain enough Vδ1+T cells to observe any clonal expansion (perhaps not surprising given that spleen Vδ1+T cell numbers are often low in healthy donors even if frequencies are high [43]. As organoid models develop to support longer term culture, future experiments could investigate repeated malaria exposure and/or single exposure in cells from donors previously exposed to malaria in order to determine if our observed clonal expansion of Vδ2+T cells is similarly related to a primary infection or not.

Tonsil and spleen organoids provide a novel way to study human adaptive responses, including primary responses, such as to malaria. However, the organoids, as well as the use of schizont stage iRBCs, do also have limitations including sample size, donor variability, and limits to the culturing time. It is possible that if cultures could be maintained longer that differences in other T and B cell subsets or plasmablasts may have been observed following iRBC stimulation, as was observed in response to influenza [22]. Additionally, these limitations could have influenced our observation that iRBC exposure and γδ T cell depletion do not impact antibody responses to LAIV. Future studies should dissect the contribution of different parasite components (including hemozoin, a byproduct of iRBC lysis), test the addition of other parasite stages including merozoites, and examine any impact of depleting γδ T cells at timepoints later than D7. Given the public health impact of reduced immune responses to vaccines among malaria-endemic populations, it will be important for future studies to investigate the possible link between Vδ2+T cell responses *to Pf* and impaired antibody responses to both malarial and non-malarial antigens.

Together, our results provide important insights into the phenotype of Vδ2+T cells responding to malaria, as well as to the strengths of using tonsil and spleen organoids to study innate-adaptive interactions in the immune response to malaria. This work has important implications for our understanding of the role for Vδ2+T cells in malaria and provides justification for incorporating approaches targeting these cells into future malaria therapeutics.

## Methods

### Ethics

Written informed consent was obtained from all participant or legal representatives. Ethics approval for malaria-naïve samples was obtained from Stanford University's Human and Animal Research Subjects Compliance Office (Stanford IRB-60741 and IRB-30387). Ethics approval for the use of human samples in the Ugandan studies was obtained from the Mbale Regional Referral Hospital research ethics committee (MRRH-2022–235) and Alfred Human Research and Ethics Committee for the Burnet Institute (#284/23).

### Preparation of frozen *Plasmodium falciparum* iRBCs

3D7 *Plasmodium falciparum* parasites were cultured in standard conditions and purified at schizont stages by MACS purification (Miltenyi Biotec, Gaithersburg, MD). Malaria-infected red blood cells (iRBCs) or uninfected red blood cells

(uRBCs) were frozen in 25 µl human serum or fetal bovine serum (FBS) and 3x the volume of the uRBC/iRBC pellet of glycerolyte to the pellet. To thaw iRBCs/uRBCs for stimulation experiments, 20 µl of 3.5% NaCl was added to each tube and rested for 5 minutes. 1mL of the same solution was added, and the sample was centrifuged and the supernatant removed. After scraping the tube along the biosafety cabinet grate to dislodge parasites, the iRBCs/uRBCs were washed in 1.8% NaCl and then PBS. Parasites were counted using a hemocytometer.

## Tissue processing and organoid culture

Tonsil samples (Table 1) were obtained from Stanford Hospital from adult and pediatric patients undergoing tonsillectomies. All patients or legal representatives gave written informed consent (Stanford IRB-60741 and IRB-30387). Tonsil samples were also obtained from Jinja Hospital, Uganda, from pediatric patients undergoing tonsillectomies. Informed consent was given by parents and guardians (MRRH-2022–235). Tonsil tissues were processed to obtain single-cell suspensions as previously described [22]. Briefly, tonsil samples were collected immediately after surgery and incubated in Ham's F12 medium (Gibco) containing Normocin (Invivogen) and Penicillin/Streptomycin for at least 30min at 4C before processing. Afterwards, tonsils were manually minced into small pieces (~5 mm x 5 mm x 5 mm) and passed through a 100um filter using a syringe plunger to obtain a cell suspension. Filter was repeatedly washed with complete immune organoid medium (RPMI with glutamax, 10% FBS, 1X Penicillin/Streptomycin/Amphotericin B, 1mL Normocin, 5mL Insulin/transferrin/selenium supplement, 1X non-essential amino acids, 1X sodium pyruvate). Debris was removed by Ficoll density gradient separation for Stanford but not Ugandan samples. Collected cells were washed with PBS, counted, and frozen in cryovials in FBS + 10% DMSO. Frozen aliquots were stored at -180°C until use.

Spleen samples (Table 1) were obtained from adult organ donors through Donor Network West (DNW). Spleen tissue was incubated in Ham's F-12 medium supplemented with Normocin and Penicillin-Streptomycin for at least 30 minutes at 4C. Spleen tissue was cleaned up by removing large blood vessels, fat, and outer capsule, sectioned into ~5g pieces, manually minced into a paste, and resuspended in 10ml of dissociation media per section (complete RPMI (RPMI1640 with Glutamax, 10% fetal bovine serum (FBS), 1x Penicillin-Streptomycin, 1x Non-Essential Amino Acids, 1x Sodium Pyruvate, 1x Insulin-Transferrin-Selenium, and 1x Normocin) supplemented with 1 mg/ml collagenase IV and 200U/ml DNase I). Spleen suspension was transferred to gentleMACS C-tubes, incubated at 37C and dissociated by 2x mechanical disruption using gentleMACS Dissociator (protocol: Spleen_04_01). Dissociated spleen was passed through a 100um filter, resuspended in PBS supplemented with 2% FBS and 6mM EDTA, and washed once. Red blood cells were lysed by resuspending in ACK lysis buffer for 5 minutes at room temperature. Granulocytes were removed using EasySep Direct Human PBMC Isolation Kit (StemCell Technologies) per manufacturer's instructions. Cells were counted, and frozen in cryovials in FBS + 10% DMSO. Frozen aliquots were stored at -180°C until use.

Cryopreserved tonsil or spleen cells were thawed and counted, 6 x 10^6 cells in 100 µl immune organoid media were added to each transwell (Millipore PICM01250) in 12-well plates. Immune organoid media containing 0.5 µl g/mL recombinant human B cell activation factor (BAFF, Biolegend 559608) was added to the outside of the well and uRBC/iRBC were added to the transwell at a ratio of 2:1 (i.e., 12 x 10^6 iRBC or uRBC/well). Multiple ratios of iRBCs:cells were tested before choosing this ratio. For conditions containing LAIV, 1:1,000 (Flumist Quadrivalent, AztraZeneca, obtained through Stanford Pharmacy) was added directly on top of each transwell. Experiments comparing addition of LAIV at day 0 vs. day

**Table 1. Malaria-naïve tonsil and spleen donors.**

|  | Tonsil (n = 17) | Spleen (n = 4) |
|---|---|---|
| Age, years, median [IQR] | 12 [7-19] | 30.5 [28-34] |
| Sex, male | 11 (64.71%) | 4 (100%) |

3 indicated that LAIV-specific responses were higher when LAIV was added at day 0, so the majority of experiments utilized this timepoint. Fresh media containing BAFF was added to the outer well every 3–4 days. Supernatants and immune organoid cells were collected at day 7 and 14 for tonsils and day 14 and 21 or 28 for spleens. For γδ T cell depletion experiments, tonsil or spleen single-cell suspensions were washed in MACS buffer (PBS containing 10% FBS and 0.5M EDTA) and γδ T cells were isolated using a negative purification kit from Miltenyi Biotec.

**Flow cytometry using malaria-naïve cells**

Cells were transferred to Eppendorf tubes and FACS Buffer (PBS containing 0.4% EDTA and 0.5% Bovine Serum Albumin) was used to wash all cells from inside the transwell. Cells were spun down, washed in FACS Buffer and counted. 250,000 cells/well in PBS were plated in each of 96-well U-bottom plates (for 3 different flow cytometry panels) and spun down. For the T cell panel (Table 2), 50 µl of anti-CCR6 (1:50 in PBS) was added and incubated at 37$^{\circ}$C for 30 minutes. Cells were washed in FACS buffer and re-suspended in cocktail of all other surface antibodies (Table 2) at room temperature for 30 minutes. Cells were washed twice, re-suspended in PBS containing 2% paraformaldehyde (company), and stored overnight at 4$^{\circ}$C. For the B cell (Table 3) and innate cell (Table 4) panels, the same steps for surface antibody staining, washes, and storage were followed.

For the experiment in S1H Fig characterizing Vδ2 + T cells, tonsil organoids from 3 donors were stimulated for 14 days. At day 7 and day 14, cells were spun down, washed with FACS buffer, and then re-suspended in a cocktail of surface antibodies (including Vδ2 + T cell activation and memory markers) (Table 5) for 30 minutes at room temperature. Cells were washed twice and re-suspended in PBS.

For all experiments mentioned above, FMO controls were prepared for all markers that did not have distinct positive and negative populations. Samples were run on a Attune Nxt Cytometer (Thermo Fisher Scientific) and data was analyzed using FlowJo (Treestar, Inc). Compensation was performed using UltraComp beads (ThermoFisher Scientific) with single stains added. Differences in cell frequencies between conditions or timepoints were analyzed using the paired Wilcoxon rank-sum test.

**Flow cytometry on Ugandan tonsils to characterize Vδ2 + T cells**

Cryopreserved tonsil cells from Ugandan donors (Table 6) were thawed and counted in complete medium (RPMI 1640 with L-Glutamine, 25mM HEPES, 10% FBS, 50 U/mL Benzonase and 1x Penicillin-Streptomycin). A total of $2 \times 10^6$ cells were plated in a 96-well round-bottom plate (Corning) and rested overnight at 37$^{\circ}$C. Following resting, the cells were

**Table 2. T cell panel.**

| Antibody | Fluorophore | Channel | Company | Clone | Catalog # | Volume per 50 µl | FMO? |
|----------|-------------|---------|---------|-------|-----------|------------------|------|
| CCR6 | AlexaFluor700 | R2 | BioLegend | G034E3 | 353434 | 1 | X |
| Live/dead | Aqua | V2 | Invitrogen | – | L34965 | 0.02 | |
| CD14 - Dump | BV510 | V2 | BioLegend | M5E2 | 301842 | 0.16 | |
| CD19 - Dump | BV510 | V2 | BioLegend | HIB19 | 302242 | 0.16 | |
| CD3 | FITC | B1 | BioLegend | SK7 | 344804 | 1 | |
| CD4 | BV650 | V4 | BioLegend | OKT4 | 317435 | 0.3 | |
| CD8 | BV605 | V3 | BD | SK1 | 564116 | 1 | |
| CXCR5 | BV711 | V5 | BioLegend | J252D4 | 356934 | 0.6 | X |
| PD-1 | PE | B2 | BD | EH12.1 | 560795 | 3 | X |
| ICOS | APC-Cy7 | R3 | BioLegend | C3984A | 313529 | 0.8 | X |
| CXCR3 | BV421 | V1 | BD | 1C6/CXCR3 | 562558 | 1.6 | X |
| CD45RA | PerCp-Cy5.5 | B3 | BD | HI100 | 563429 | 1 | X |
| CCR7 | APC | R1 | BioLegend | G043H7 | 353214 | 1 | X |

**Table 3. B cell panel.**

| Antibody | Fluorophore | Channel | Company | Clone | Catalog # | Volume per 50 µl | FMO? |
|---|---|---|---|---|---|---|---|
| Live/dead | Aqua | V2 | Invitrogen | – | L34965 | 0.02 | |
| CD3 - Dump | BV510 | V2 | BioLegend | SK7 | 344828 | 0.25 | |
| CD14 - Dump | BV510 | V2 | BioLegend | M5E2 | 301842 | 0.16 | |
| IgD | BV785 | V6 | BioLegend | IA6–2 | 348241 | 0.2 | X |
| IgG | FITC | B1 | BD | G18-145 | 555786 | 0.5 | X |
| IgM | PerCP-Cy5.5 | B3 | BD | G20-127 | 561285 | 0.3 | X |
| CD19 | PE | B2 | BioLegend | HIB19 | 302207 | 0.1 | |
| CD20 | APC-Cy7 | R3 | BioLegend | 2H7 | 302314 | 0.16 | |
| CD21 | APC | R1 | BD | B-LY4 | 559867 | 0.16 | |
| CD27 | BV421 | V1 | BD | M-T271 | 562513 | 0.66 | X |
| CD38 | Alexa700 | R2 | BD | HIT2 | 560676 | 3 | X |
| CD138 | BV605 | V3 | BioLegend | MI15 | 356519 | 1 | X |
| CD95 | BV711 | V5 | BioLegend | DX2 | 305644 | 1 | X |

**Table 4. Innate cell panel.**

| Antibody | Fluorophore | Channel | Company | Clone | Catalog # | Volume per 50 µl | FMO? |
|---|---|---|---|---|---|---|---|
| Live/dead | Aqua | V2 | Invitrogen | – | L34965 | 0.02 | |
| CD19 - Dump | BV510 | V2 | BioLegend | HIB19 | 302242 | 0.16 | |
| VD1 | APC | R1 | Invitrogen | TS8.2 | 17567942 | 1 | |
| VD2 | PerCP | B3 | BD | B6 | 331410 | 1 | |
| CD3 | FITC | B1 | BioLegend | SK7 | 344804 | 1 | |
| CD14 | APC-H7 | R3 | BioLegend | M5E2 | 561384 | 1 | X |
| CD56 | BV711 | V5 | BioLegend | HCD56 | 318336 | 1 | X |
| CD16 | BV785 | V6 | BioLegend | 3G8 | 302046 | 1 | X |
| CD123 | BV650 | V4 | BioLegend | 6H6 | 306020 | 1 | X |
| CD11c | Alexa700 | R2 | BioLegend | 3.9 | 301648 | 1 | X |
| HLA-DR | BV605 | V3 | BioLegend | L243 | 307640 | 1 | X |
| CD127 | PE | B2 | BD | HIL-7R-M21 | 561028 | 1 | X |

simultaneously stained with pre-stimulation antibody panel (**Table 7**) and stimulated with 25ng/mL of Phorbol 12-Myristate 13-Acetate (PMA; Sigma) and 1 µg/mL of Ionomycin (Sigma) at 37°C. Brefeldin A (BD Biosciences) and Monensin (BD Biosciences) was added after 30 mins of stimulation. The cells were stimulated for an additional 3.5 hours. Following stimulation, cells were washed and stained with Chemokine antibody panel (**Table 7**) at 37°C for 45 mins. Viability staining with ViaDye Red at 1:10,000 dilution (Cytek) and human Fc Block (BD Biosciences) were then added in 1x PBS. The cells were then stained with Surface antibody panel (Table 7) for 20 mins at RT before a fixation and permeabilization process using the eBioscience Fixation/Permeabilization kit (Thermo Fisher). Finally, the cells were stained with Intracellular antibody panel (Table 7) for 30 mins at 4°C and stabilized using Stabilizing Fixative (BD Biosciences). Cell acquisition was performed using the Aurora (Cytek) flow cytometer.

## Peripheral PCR for malaria parasites in Ugandan cohort

Whole blood samples were collected onto Whatman 3 mm chromatographic filter paper and air-dried at room temperature. Dried blood spots (DBS) were cut using a sterile hole punch and placed into a 1.5 mL microcentrifuge tubes. To lyse

 

**Table 5. Vδ2＋T cell Panel for experiment in S1H Fig.**

| Antibody | Fluorophore | Channel | Company | Clone | Catalog # | Volume per 50 µl | FMO? |
|---|---|---|---|---|---|---|---|
| CD57 | BV421 | V1 | BD | NK-1 | 563896 | 0.5 | X |
| Live/dead | Aqua | V2 | Invitrogen | – | L34965 | 0.13 | |
| CD19 - Dump | BV510 | V2 | BioLegend | HIB19 | 302242 | 0.25 | |
| CD14 - Dump | BV510 | V2 | BioLegend | M5E2 | 301842 | 0.25 | X |
| TIM-3 | BV605 | V3 | BioLegend | F38-2E2 | 345018 | 0.5 | |
| CD45RA | BV650 | V4 | BioLegend | HI100 | 304136 | 0.5 | X |
| CD16 | BV711 | V5 | BioLegend | 3G8 | 302044 | 0.5 | X |
| CD8 | BV785 | V6 | BioLegend | RPA-T8 | 301046 | 0.5 | X |
| CD27 | FITC | B1 | BioLegend | M-T271 | 356404 | 0.5 | X |
| VD2 | PE | B2 | BioLegend | B6 | 331408 | 0.5 | |
| CD69 | PerCP-Cy5.5 | B3 | BioLegend | FN50 | 310926 | 0.5 | X |
| HLA-DR | APC | R1 | BioLegend | L243 | 307610 | 0.5 | X |
| CD3 | APC-H7 | R3 | BD | SK7 | 560176 | 1 | |

**Table 6. Clinical characteristics of malaria-endemic children.**

| | Uninfected (n＝10) | Infected (n＝10) |
|---|---|---|
| Age, years, median [IQR] | 5.5 [2.25–9.25] | 6.0 [3.7–7.0] |
| Sex, male | 5 (50%) | 5 (50%) |
| Reason for tonsillectomy | | |
| Sleep Apnea | 3 (30%) | 4 (40%) |
| Hypertrophy | 1 (10%) | 0 (0%) |
| Both | 6 (60%) | 6 (60%) |
| Peripheral blood parasitemia (Parasites/µL) | 0 | 0.16 [0.05–0.45] |

red blood cells, 1 mL of 0.5% saponin was added to each tube and incubated overnight at 4°C. The samples were subsequently washed with 1×PBS. DNA extraction was performed using Chelex resin. 150 µL of 6.7% Chelex resin solution, prepared in nuclease-free water, was added to each tube. Tubes were incubated at 95°C for 10 minutes in a heat block, followed by centrifugation. The resulting supernatant containing genomic DNA, was carefully transferred to a new tube and stored for downstream PCR analysis. Malaria parasites were quantified using quantitative reverse transcription polymerase chain reaction (qPCR) targeting the varATS region. The samples and controls were analyzed with the TaqMan Gene Expression Master Mix (Thermo Fisher, catalog number 4369016), TaqMan Probe (5'-3' sequence: 6-FAM-trttccataaatggt-NFQ-MGB), forward primers (5'-3': cccatacacaaccaaytggaas), and reverse primers (5'-3': Ttcgcacatatctctatgtctatct), and 10X TE Buffer (Teknova, catalog number T3457), following the manufacturers' instructions. The qPCR cycling conditions were set as follows: Pre-incubation at 50°C for 2 minutes; Initial denaturation at 95°C for 10 minutes; Denaturation at 95°C for 15 seconds; and Annealing & Elongation at 55°C for 1 minute and Number of cycles to 60 cycles.

## ELISA

Cell culture supernatants were stored at -80°C and thawed at 4°C overnight before an experiment. For flu protein ELISAs, high-binding, flat-bottom ELISA plates (Fisher Scientific) were incubated overnight at 4°C with 50 µL of 2 µg/mL HA protein (Sino Biological HA Recombinant Influenza a Virus Protein, Subtype H1N1 (A/California/04/2009), His Tag) in ELISA Bicarbonate Buffer. The following day, the plate was washed 3 times with wash buffer (PBS＋0.05% Tween20). 200 µL

**Table 7. Antibody Panels for Cytokine Induction Assay using Ugandan Samples.**

| Antibody | Fluorophore | Channel | Company | Clone | Catalog # | Dilution |
|---|---|---|---|---|---|---|
| **Pre-Stimulation Antibody Panel** | | | | | | |
| CD107a | BUV395 | UV2 | BD Biosciences | H4A3 | 565113 | 1:25 |
| **Chemokine Antibody Panel** | | | | | | |
| TCR γδ | BV421 | V1 | BioLegend, Inc. | B1 | 331218 | 2:50 |
| **Surface Antibody Panel** | | | | | | |
| CD38 | BUV737 | UV14 | BD Biosciences | HB7 | 564686 | 1:200 |
| CD3 | BUV805 | UV16 | BD Biosciences | SK7 | 612893 | 1:250 |
| CD27 | Pacific Blue | V3 | BioLegend, Inc. | M-T271 | 356413 | 1:200 |
| Vδ2 | BV650 | V11 | BD Biosciences | B6 | 743752 | 1:50 |
| CD45RA | BV711 | V13 | BioLegend, Inc. | HI100 | 304137 | 1:100 |
| **Intracellular Antibody Panel** | | | | | | |
| IFNγ | BV605 | V10 | BD Biosciences | B27 | 562974 | 1:50 |
| TNF | BV750 | V14 | BD Biosciences | Mab11 | 566359 | 1:100 |
| Granzyme B | AF700 | R4 | BD Biosciences | GB11 | 560213 | 1:500 |

blocking solution (1% BSA in PBS) was added and incubated for 2 hours at room temperature. After 2 washes, primary antibody (neat, 1:10, 1:100, 1:300; 100 µL/well) re-suspended in PBS was added and incubated for 1 hour at room temperature. After 5 washes, secondary antibody (Anti-human Fc-IgG-HRP, Southern Biotech, 1:50,000, 100 µL/well) in wash buffer was added and incubated for 1 hour at room temperature. After 5 washes, TMB Substrate (Thermo Scientific 1-Step Ultra TMB-ELISA, 100 µL/well) was added and incubated for 10 minutes in the dark. The reaction was stopped by adding 100 µL/well 1M $H_2SO_4$. The plate was read at 450 nm absorbance. Area under the curve (AUC) was calculated using Graphpad Prism for the curve under the 4 dilutions tested, and differences in AUC between conditions or timepoints were analyzed using the paired Wilcoxon rank-sum test.

ELISAs for malaria-specific responses were performed using protocols from the Boyle lab (QIMR Berghofer). Schizont lysate was prepared by magnetically purifying schizont-stage parasites (and uRBCs) as described above and re-suspending pellets at 20% hematocrit in PBS mixed with protease inhibitor cocktail (Sigma). 250 µL of re-suspended cells was aliquoted into tubes, freeze-thawed for 5 cycles, and sonicated for 5 cycles. Samples were then spun down and supernatants were collected, pooled, aliquoted, and stored at -80°C. Schizont lysate (1:500) and parasite antigens (MSP1/AMA1/ MSP2, 0.5 µg/mL, kindly provided by James Beeson and Patrick Duffy) were prepared in PBS and incubated overnight at 4°C on Nunc ELISA plates. The following day, plates were washed 3 times with wash buffer and incubated in blocking buffer (1% casein) for 2 hours at 37°C. Organoid supernatants were prepared at different dilutions in antibody buffer (0.1% casein) and added to the plates for 1 hour at room temperature. Plates were washed 3 times and then secondary antibody (suspended in antibody buffer) was added for 1 hour at room temperature. After 3 washes, TMB Substrate was added for 15 minutes in the dark. The reaction was stopped using 1M $H_2SO_4$ and the plate was read at 450 nm absorbance.

## Sample preparation for single cell multiomics analysis using the BD Rhapsody platform

Spleens from 3 donors were used and an aliquot of cells from d0 was frozen for each. Organoids were set up and stimulated with uRBC or iRBC (as described above) for 10 days with media changes every 3–4 days. Stimulated cells, as well as d0 cells, were then stained with an anti-Vδ2 + antibody (PerCP-conjugated, 1:50) and a cell viability dye (Live/Dead Aqua, 1:400) for 30 minutes, washed, and then sorted on a Sony cell sorter. For each sample, 8,000–40,000 Vδ2 + γδ T cells were sorted and depending on the number collected, 10,000–30,000 Vδ2- cells were sorted in order to reach a total of 40,000–50,000 cells per sample. Cells were sorted into cold PBS containing 2% fetal bovine serum. Fewer cells were

collected for corresponding PBMC samples, which as would be expected from a blood sample following autopsy, had a lot of dead cells and debris. The percentage of total cells that were Vδ2 + T cells ranged from 20-80% per sample (80% for all iRBC-stimulated cells, lower for some uRBC-stimulated and d0).

## Single cell multiomics analysis using the BD Rhapsody platform

Samples were re-suspended in a staining buffer at 200–800 cell/µL in ~500ul and transferred to the Stanford Human Immune Monitoring (HIMC) Core. Samples were stained with oligonucleotide-conjugated Sample Tags using the BD Human Single-Cell Multiplexing Kit (Cat. # 633781) in BD stain buffer following the manufacturer protocol. Barcoded samples were then washed and spun down at 350xg for 10 minutes and pooled. Pooled sample was then stained concurrently with an oligonucleotide-conjugated antibody cocktail (Table 8). Stain was in BD stain buffer for 30 minutes on ice, and samples were then spun down at 350xg for 10 minutes and washed three times. Pellet was resuspended in the Rhapsody buffer for capture using BD Rhapsody Enhanced Cartridge Reagent Kit V3 (Cat. #667052). Cell capture and library preparation were completed using the BD Rhapsody WTA Amplification Kit (Cat. 633801). Briefly, cells (~25,000 live cells per sample were captured with beads in microwell plate, followed by cell lysis, bead retrieval, cDNA synthesis, template switching and Klenow extension, and library preparation in the HIMC core, following the BD Rhapsody protocol. Libraries were prepared for T cell receptor, sample tags, Abseq, and whole transcriptome using BD Rhapsody Human TCR/BCR Amplification Kit (Cat. #667400). Sequencing was completed on NovaSeq XP (Illumina, San Diego, CA) at Novogene (Sacramento, CA). Rhapsody data were processed using BD Rhapsody sequence analysis pipeline v2.0 hosted in the Seven Bridges Genomics cloud platform (now Velsera. San Francisco, CA) and the primary pipeline analysis results with V(D)J processing incorporated to generate all output files, including H5AD and SEURAT data format for secondary analysis and visualization.

## Single cell transcriptomics and TCR analysis

Data was imported into Seurat (4.3). DoubletFinder 2.0.4 [44] was used to remove doublets using a target rate of 26%. This doublet rate was inferred by the frequency of cells that had multiple hashing antibodies as well as the automated visual inspection of the microwells performed by the instrument. Additionally, we excluded cells with fewer than 200 or more than 3000 genes or more than 10% mitochondrial genes. Cells with ambiguous or multiple hashing antibody sequences were removed. CITE-Seq read counts were normalized using centered log ratio transformation, while mRNA

**Table 8. AbSeq Antibodies.**

| Catalog number | Product description |
|---|---|
| 940000 | Hu CD3 Olgo AHS0033 SK7 25Tst |
| 940003 | Hu CD8 Olgo AHS0027 RPA-T8 25Tst |
| 940006 | Hu CD16 Ab-O AHS0053 3G8 25Tst |
| 940010 | Hu HLA-DR Olgo AHS0035 G46-6 25Tst |
| 940011 | Hu CD45RA Ab-O AHS0009 HI100 25Tst |
| 940012 | Hu CD127 Olgo AHS0028 HIL-7R-M21 25Tst |
| 940013 | Hu CD38 Ab-O AHS0022 HIT2 25Tst |
| 940251 | Hu CD44 Olgo AHS0167 L178 25Tst |
| 940295 | Hu V GMA 9 TCR Olgo AHS0218 B3 25Tst |
| 940319 | Hu CD27 Olgo AHS0249 L128 25Tst |
| 940394 | HU CCR7(CD197) Olgo AHS0273 2-L1-A 25Tst |
| 940066 | Hu TIM-3 Olgo AHS0016 7D3 25Tst |
| 940232 | Hu CD122 Olgo AHS0146 MIK-BETA3 25Tst |

gene counts were normalized separately using standard log normalization. Expression levels were then scaled and centered before variable feature selection. The 2500 most variable genes were used as input to compute principal components. Harmony (version 1.2.3) was used to reduce the impact of individual variation on clustering and dimensionality reduction. Based on Elbow plot inspection, the first 12 harmony "principal components" were used to compute UMAP and to cluster cells using the Seurat implementation of the Louvain algorithm (resolution 0.8). Differential gene expression was performed using the Seurat implementation of the non-parametric Wilcoxon rank sum test. Gene set enrichment was performed with clusterProfiler (4.6.2) using fold change ordered gene lists as input. Biological Process GO terms were queried using the org.Hs.e.g.,db (3.16.0). TCR repertoires were analyzed using scRepertoire (1.8.0) using amino acid CDR3 sequences to group clones. For clonal analyses we retained only cells that had high confidence VDJ genes called that matched their ascribed transcriptional identity as either αβ or γδ T cell, respectively. Single cell RNA/TCR sequencing data are available at DOI: 10.5061/dryad.3r2280gw3 [45].

## Supporting information

**S1 Table. Frequency changes of top 50 clones CDR3 sequences.**
(CSV)

**S1 Fig. Germinal centers form in organoids in response to iRBC stimulation A.** Microscope image showing formation of germinal centers in tonsil organoid stimulated with iRBC. B. Cell frequencies (of all single cells) at day 21/28 (1–3 donors depending on cell type). C. Gating strategy for B cell panel. D. Gating strategy for T cell panel. E. Gating strategy for innate cell panel. F. Frequencies of cells that do not change in tonsil organoids in response to iRBC (compared to uRBC). N = 10. Bars indicate medians. P-values were calculated using the Wilcoxon ranked sum test with correction for multiple comparisons using Benjamini-Hochberg method (FDR 0.05). G. Expression of HLA-DR on Vδ2 + T cells from 14 tonsil donors with both D7 and D14 timepoints, and 3 additional donors with only D7. Bars indicate medians. H. Tonsil organoids from 3 donors were stimulated for 14 days and a select set of Vδ2 + T cell surface markers were evaluated. Frequencies of Vδ2 + T cell subsets expressing Tim-3 and CD8, and frequencies or of central memory and effector memory subsets were assessed at multiple timepoints and conditions. Bars indicate medians.
(DOCX)

**S2 Fig. AbSeq and RNA expression defines cell clusters.** A. Expression of markers defining all cell clusters. Numbers correspond to clusters in Fig 2B. B. Expression of antibody surface markers across all clusters. C. Expression of TCR gamma chain genes (left) and delta chain genes (right) across cell clusters. D. Expression of class II antigen presentation markers across clusters. E. UMAP showing distribution of select genes associated with cell division that are specifically enriched in Vδ2 + cells after iRBC stimulation. See Fig 2B for the location of γδ T cell clusters.
(DOCX)

**S3 Fig. Receptor repertoires of conventional T cells in the spleen organoid are relatively stable between experimental conditions.** A. Bar graph showing the percentage of each donor's conventional T cell receptor repertoire occupied by clones of different orders of magnitude. B. Alluvial plot showing the distribution of the 10 largest conventional T cell clones per sample. Only cells with a high confidence α and β chain were included.
(DOCX)

**S4 Fig. Changes in Vδ2- T cell phenotypes in malaria infected children Isolated Tonsillar mononuclear cells from malaria-endemic Ugandan children with asymptomatic parasitemia (infected) or no Pf infection (uninfected) were stimulated with PMA and Ionomycin for 24 hours.** A. Vδ2- frequency of total T cells. B. Frequency of Vδ2- T effector memory (TEM: CD27-CD45RA-), T central memory (TCM: CD27 + CD45RA-), T terminally differentiated effector memory (TEMRA: CD27-CD45RA+) and T *naive* (TNAIVE: CD27 + CD45RA+). C. Frequency of Vδ2- T

cell intracellular Granzyme-B expression. Comparisons performed by Mann-Whitney U test. Center line representing the median, box limits indicating the upper and lower quartiles, whiskers extending to 1.5 times the interquartile range. D. Pie chart depicts co-expression of CD38, CD107a, Granzyme-B, IFNγ and TNF by Vδ2- T cells. Comparisons performed by permutation test. E. Frequency of $CD38^+CD107a^+Granzyme-B^+IFN\gamma^+TNF^+$ and $CD38^+CD107a^+Granzyme-B^-IFN\gamma^+TNF^+$ Vδ2- T cells. Bar representing mean and whiskers extend to standard deviation. Comparisons performed by Mann-Whitney U test.
(DOCX)

**S5 Fig. iRBC exposure does not impact cellular frequencies following LAIV vaccine.** T cell subsets, B cell subsets, and most innate cell frequencies, whether of all single cells (A) or of parent gate (B), are not different following stimulation with iRBC+LAIV compared to the uRBC+LAIV or LAIV alone conditions. LAIV was added at day 0. N = 3. C. Vδ2+T cell frequencies are significantly different between iRBC+LAIV and uRBC+LAIV or LAIV alone at Dat 7 and Day 14. N = 6. D. Vδ2+T cell frequencies do not change following LAIV stimulation compared to media controls (N = 6, 3 for LAIVd0, 3 for LAIVd3).
(DOCX)

**S6 Fig. iRBC impacts antibody response to LAIV vaccine but not malaria parasites.** A. Antibody responses to flu protein from ELISAs using supernatants from cultures exposed to different stimulation conditions at Day 7, Day 14, or Day 21/28. Lines represent unique donors. Tonsil donors are indicated in black and spleens in red. Area under the curve (AUC) was calculated using the curve of absorbances across the 4 dilutions tested (neat, 1:10, 1:100, 1:300). B. Example curve of absorbances across 4 dilutions that was used to calculate AUC. C. IgG antibody response to malaria schizont extract lysate from ELISAs using undiluted supernatants from organoid cultures exposed to different stimulation conditions. Ugandan plasma was added at 1:1000. D. IgG antibody response to malaria parasite antigens (MSP1, AMA1, MSP2) from ELISAs using supernatants (1:10 or 1:100) from organoid cultures exposed to different stimulation conditions. Ugandan plasma was added at 1:1000.E. IgM antibody response to malaria schizont extract lysate from ELISAs using supernatants from unstimulated, uRBC-stimulated or iRBC-stimulated cultures at day 7 (n = 13 tonsils and 1 spleen). Area under the curve (AUC) was calculated using the curve of absorbances across the 4 dilutions tested (neat, 1:10, 1:100, 1:300).F. IgM antibody response to malaria schizont extract lysate from ELISAs using undiluted supernatants from unstimulated vs. iRBC-stimulated organoid cultures (n = 14 tonsils and 3 spleens) at days 7 and 14. Absorbance values are average of 2 replicates.
(DOCX)

**S7 Fig. γδ+T cell depletion does not impact immune responses at day 7.** Experiments were performed in which γδ+T cells were depleted from organoids prior to 7-day stimulation with uRBC/iRBC +/- LAIV. Effectiveness of Vδ2+ and Vδ1+depletion is shown in A. T cell responses (B), B cell responses C), and innate cell responses (D) are no different between conditions containing γδ+T cells or depleted of γδ+T cells.
(DOCX)

**S1 Data. Raw experimental data related to Figs 1, 4, 5, S1, S4, S5, S6, and S7.**
(ZIP)

## Acknowledgments

We acknowledge Dr. Taia Wang (Stanford University) and her lab members for allowing us to use their plate reader and cell sorter, Jason Nideffer (Jagannathan lab, Stanford) for providing training on the sorter, and members of Michelle Boyle's lab (QIMR Berghofer) for providing assistance with their schizont ELISA protocols. We are immensely grateful to all of the patients and families who provided tonsil and spleen material for these studies, without whom none of this

work would be possible. We acknowledge the traditional custodians of the lands where our laboratories are located: the Muwekma Ohlone Tribe (Stanford, California) and the Boonwurrung and Wurundjeri Woi-wurrong people of the Kulin Nation (Melbourne, Australia).

## Author contributions

**Conceptualization:** Kathleen D. Press, Elsa Sola, Chloe Kashiwagi, Mark M. Davis, Prasanna Jagannathan.

**Data curation:** Kathleen D. Press, Florian Bach, Elsa Sola, Prasanna Jagannathan.

**Formal analysis:** Kathleen D. Press, Florian Bach, Prasanna Jagannathan.

**Funding acquisition:** Kathleen D. Press, Michelle Boyle, Mark M. Davis.

**Investigation:** Kathleen D. Press, Florian Bach, Elsa Sola, Kylie Camanag, Nicholas L. Dooley, Anselma Ivanawati, Damian Oyong, Mayimuna Nalubega, Abel Kakuru, Sedrack Matsiko, Felistas Nankya, Kenneth Musinguzi, Annet Nalwoga, Evelyn Nansubuga, John Ategeka, Charles Ebusu, Bakar Odongo, Chloe Kashiwagi, Xuhuai Ji, Molly Miranda, Joselyn Tachiwa-Appiah, Kareena Sandhu, Lilit Kamalyan, Kattria van der Ploeg, Michelle Boyle.

**Methodology:** Kathleen D. Press, Florian Bach, Elsa Sola, Nicholas L. Dooley, Damian Oyong, Xuhuai Ji, Kattria van der Ploeg, Michelle Boyle, Lisa E. Wagar, Prasanna Jagannathan.

**Project administration:** Kathleen D. Press, Abel Kakuru, Michelle Boyle, Mark M. Davis, Prasanna Jagannathan.

**Resources:** Michelle Boyle, Lisa E. Wagar, Mark M. Davis.

**Supervision:** Elsa Sola, Kattria van der Ploeg.

**Validation:** Prasanna Jagannathan.

**Visualization:** Florian Bach.

**Writing – original draft:** Kathleen D. Press.

**Writing – review & editing:** Kathleen D. Press, Florian Bach, Elsa Sola, Kylie Camanag, Nicholas L. Dooley, Anselma Ivanawati, Damian Oyong, Mayimuna Nalubega, Abel Kakuru, Sedrack Matsiko, Felistas Nankya, Kenneth Musinguzi, Annet Nalwoga, Evelyn Nansubuga, John Ategeka, Charles Ebusu, Bakar Odongo, Chloe Kashiwagi, Xuhuai Ji, Molly Miranda, Joselyn Tachiwa-Appiah, Kareena Sandhu, Lilit Kamalyan, Kattria van der Ploeg, Michelle Boyle, Lisa E Wagar, Mark M. Davis, Prasanna Jagannathan.

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
