## [Decision Letter · Decision Letter 0]

27 Nov 2025

Cytotoxic Vδ2+ T cell subsets expand in response to malaria in human tonsil and spleen organoids

PLOS Pathogens

Dear Dr. Jagannathan,

Thank you for submitting your manuscript to PLOS Pathogens. After careful consideration, we feel that it has merit but does not fully meet PLOS Pathogens's publication criteria as it currently stands. Therefore, we invite you to submit a revised version of the manuscript that addresses the points raised during the review process.

We look forward to receiving your revised manuscript.

Kind regards,

Steven M. Singer

Guest Editor

PLOS Pathogens

Dominique Soldati-Favre

Section Editor

Editor-in-Chief

PLOS Pathogens

orcid.org/0000-0003-2946-9497

Editor-in-Chief

PLOS Pathogens

orcid.org/0000-0002-7699-2064

**Journal Requirements:**

At this stage, the following Authors/Authors require contributions: Florian Bach, Elsa Sola, Kylie Camanag, Nicholas Dooley, Anselma Ivanawati, Damian Oyong, Mayimuna Nalubega, Abel Kakuru, Sedrack Matsiko, Felistas Nankya, Kenneth Musinguzi, Annet Nalwoga, Evelyn Nansubuga, John Ategeka, Charles Ebusu, Bakar Odongo, Chloe Kashiwagi, Xuhuai Ji, Molly Miranda, Joselyn Tachiwa-Appiah, Kareena Sandhu, Lilit Kamalyan, Kattria van der Ploeg, Michelle Boyle, Lisa E Wagar, Mark M. Davis, and Prasanna Jagannathan. Please ensure that the full contributions of each author are acknowledged in the "Add/Edit/Remove Authors" section of our submission form.

- TM on pages: 31, and 33.

Potential Copyright Issues:

i) Figures 1A, 2A, and 5A. We note that the figures are created through BioRender. Please confirm that you hold a Premium account and provide a pdf copy of the CC BY 4.0 Licence as provided by BioRender. For instructions on how to generate a CC BY 4.0 license for your figure, please see the guidelines here: https://help.biorender.com/hc/en-gb/articles/21282341238045-Publishing-in-open-access-resources.

If you are using the free assets from BioRender, we are unable to publish these images as they are licenced under a stricter licence than CC BY 4.0. In this case we ask you to remove the BioRender images and replace them with open source alternatives.

See these open source resources you may use to replace images / clip-art:

- https://bioart.niaid.nih.gov/

- https://bioicons.com/

- https://healthicons.org/

- https://scidraw.io/

- https://reactome.org/icon-lib

- https://www.phylopic.org/images

- https://journals.plos.org/plosbiology/article?id=10.1371/journal.pbio.3002395

6)  Thank you for stating that "Single cell RNA/TCR sequencing data have been deposited in DRYAD and are available at DOI: 10.5061/dryad.3r2280gw3." Should your submission be accepted, we will require the following information in your Data Availability Statement:

1. The DOI provided by Dryad

2. The citation for your data package in the reference section of your manuscript

3. The citation for your data package in the methods section.

7) Please amend your detailed Financial Disclosure statement. This is published with the article. It must therefore be completed in full sentences and contain the exact wording you wish to be published.

1) State what role the funders took in the study. If the funders had no role in your study, please state: "The funders had no role in study design, data collection and analysis, decision to publish, or preparation of the manuscript.".

**Reviewers' Comments:**

Reviewer's Responses to Questions

**Part I - Summary**

Reviewer #1: This manuscript by Press et al, uses human tonsil and splenic organoids to examine the response to Plasmodium infected red blood cells (iRBCs) and to live attenuated influenza vaccine. After incubation over 14-28 days (depending on organoid) the major change in cell populations was the expansion of Vd2 GD T cells. This is a small population in the culture but does appear to be the only real difference observed.

The novelty of this study is that it is the first to examine responses during Plasmodium infection using tonsil / splenic cultures. I appreciate that this work is not trivial to carry out. The analysis of the GD T cells are insightful and demonstrate how organoid cultures could be used to dissect some of the responses observed in human studies. As such I can see this study would be helpful to the community in some regards. However, unfortunately I am not sure that the data supports a general utility for this approach with the current optimization / data obtained. If gamma delta cells are the primary difference how would one go about studying the other cells of the immune system with this approach? It perhaps needs more optimization.

Reviewer #2: This study by Press et al seeks to explore lymphoid tissue responses of Vdelta2 T cells (Vd2+) in tonsil and spleen organoids. They look at cellular responses of Vd2+ cells in spleen organoids in vitro and tonsils from donors from Uganda. The main findings indicate Vd2+ cells expand in organoid cultures and this is underpinned by transcriptional signatures of cytolytic effector cells and clonal expansion of the gd TCR.

Reviewer #3: Strengths of the Study:

This ex vivo model represents a relatively new and emerging tool of the lymphoid organoid system for studying malaria immunology and vaccine efficacy.

Press et al. provide a valuable human lymphoid-organoid model for studying malaria immune responses, in particular, understanding γδ T-cell biology in human lymphoid tissues during primary malaria infection.

Weaknesses / Major Concerns:

The authors should discuss their findings in greater depth and align their conclusions with the existing literature comparing non-endemic adults with endemic adults and children. The authors should clearly state that the organoid model primarily reflects a first malaria infection and cannot be extrapolated to repeated infections.

The interpretation of the Uganda children’s tonsils requires caution, since Vδ2⁺ frequencies were not increased in infected children (Fig. 4A), in contrast to the organoid cultures. Moreover, studies analysing human peripheral blood consistently report a decrease in Vδ2 T cells in chronically exposed populations (Jagannathan et al., 2014; Diallo et al., 2019). The authors should interpret the tonsillar cytotoxic phenotype in this context and extrapolate with caution and rigor to the γδ T-cell analyses in individuals from endemic regions.

The authors should also assess Vδ1 and Vδ3 γδ T-cell subsets in the Ugandan children's tonsils. These subsets are predominant in human lymphoid tissues, such as pediatric tonsils, and often expand or differentiate into effector cells during malaria infection. It is important to report, if possible, the Vδ1/Vδ3 potential contribution to the CD27⁺CD45RA⁺ T-naïve γδ population, as well as the frequencies and activation states in these children. Including these data would substantially strengthen the manuscript by providing a more complete picture of γδ T-cell repertoire in the lymphoid tissues of malaria-exposed children.

The authors claim that Vδ2⁺ activation is malaria-antigen specific by showing the expression of activation markers. However, without specific mechanistic assays (Sandstrom et al., Science, 2014; Schofield et al., BMC Med, 2017), the conclusion is indirect and should be stated more cautiously.

In the study, Vγ9Vδ2 T cells upregulated surface expression of APC markers, HLA-DR, CD74, TAP, and CIITA, in response to iRBCs. However, this expression is insufficient to attribute functional antigen processing and presentation (Brandes et al., Science, 2005; Howard et al, 2017). The authors should revise this by stating that this conclusion requires experimental validation.

**Part II – Major Issues: Key Experiments Required for Acceptance**

Please use this section to detail the key new experiments or modifications of existing experiments that should be absolutely required to validate study conclusions.required to validate study conclusions.required to validate study conclusions.required to validate study conclusions.

Reviewer #1: Required experiments: I think the data are surprising. For example, I would have expected more changes than just GD T cells. There are very few other changes observed. This makes me curious as to the applicability of the organoid system given the length of time cells were cultured. NK cells are another innate cell where there does not appear significant changes despite published literature demonstrating both responsiveness to Plasmodium iRBCs and importance in animal models. One wonders whether perhaps there is some issue with the infusion of Plasmodium iRBCs / LAIV that may need adjustment? One thing that springs to mind is that perhaps the authors should consider adding purified merozoites – whilst schizonts are immunogenic, there are several reports that point to the immunogenicity of the merozoites contained inside. Also hemozoin may be interfering with responses. What do the organoids do if merozoites are added instead of iRBCs?

The authors muse that maybe a lack of changes in antibody responses is that the cultures are not sufficiently long enough but the splenic organoids were cultured for 28 days. They mention there was little to no anti-Plasmodium antibody response but this seems odd given the likely contribution of extrafollicular responses. I would expect to see at least some IgM. The authors only show IgG.

Reviewer #2: 1. The details of the organoid cultures and validation of their 3D structure is missing (i.e. images of their structure, staining of their structure by IHC etc) and this is important for the interpretation that these findings are in organoid structures or simply co-cultures of spleen/tonsils with iRBCs?

2. The data that support the claim that Vd2 TCRs clonally expand is not clear enough to interpret. I don't know what TCR chain these figures relate to or it's not clearly in Figure legend. It's well know that the Vd2 TCR repertoire is semi invariant - Vg9 repertoires contain highly shared CDR3 sequences and often exist as presumably clonally expanded populations - however they often pair with different Vd2 TCR sequences. The authors need to show either Vd2 clonotyping or paired sequencing and more convincingly describe how this is clonal selection.

3. How do these "tissue specific" cellular and TCR responses compare to blood, it would ideal to see data on this, but failing this a detailed discussion of if these responses are an extension of observations in blood or lymphoid tissue specific?

Reviewer #3: n/a

**Part III – Minor Issues: Editorial and Data Presentation Modifications**

Reviewer #1: There are a few reported deficiencies in vaccine responses in Plasmodium infected individuals Figure 5D shows the responses to LAIV is in part for half of the samples present in the unstimulated cells showing active plasma cells. The lack of difference when LAIV is cultured in the presence of iRBCs vs uRBCs leading to the conclusion that there is likely no effect of malaria on vaccine responses should be removed – I really think the organoid system is just not sufficiently developed to make any kind of judgement on this issue based on the data shown.

Reviewer #2: The materials and methods and the description of the organoid cultures need improvement, and as mentioned more detailed and robust analysis of the TCR repertoires is required to corroborate the claims in this paper.

Figure 1 - how do the various frequencies compare to D0 spleen and tonsil samples without culture? Can this be plotted alongside the cultured control conditions.

Figure 1 - Do you have any PBMC cultures to compare these frequencies to?

Reviewer #3: The authors should include the flow cytometry gating strategy for Vδ2⁺ γδ T cells in both the organoid experiments and the Ugandan tonsil samples. In addition, the authors should clarify whether Vδ1 and Vδ3 were analyzed in the organoid system and tonsillar samples, and show the corresponding plots if available. The gating strategies for γδ, Vδ1⁺, and Vδ2⁺ subsets should be plotted and included in the main figures.

Providing a visual schematic of the Vδ2/Vδ1 dynamics in naïve, primary infection, and chronically exposed individuals would contextualize the Press et al. findings within the latest γδ T-cell biology and significantly enhance the manuscript.

PLOS authors have the option to publish the peer review history of their article (what does this mean?). If published, this will include your full peer review and any attached files.). If published, this will include your full peer review and any attached files.). If published, this will include your full peer review and any attached files.). If published, this will include your full peer review and any attached files.

...

Reviewer #1: No

Reviewer #2: No

Reviewer #3: No

**Figure resubmission:**
---

## [Decision Letter · Decision Letter 1]

17 Mar 2026

Dear Dr. Jagannathan,

We are pleased to inform you that your manuscript 'Cytotoxic Vδ2+ T cell subsets expand in response to malaria in human tonsil and spleen organoids' has been provisionally accepted for publication in PLOS Pathogens.

Best regards,

Steven M. Singer

Guest Editor

PLOS Pathogens

Dominique Soldati-Favre

Section Editor

PLOS Pathogens

Sumita Bhaduri-McIntosh

Editor-in-Chief

PLOS Pathogens

orcid.org/0000-0003-2946-9497

Michael Malim

Editor-in-Chief

PLOS Pathogens

orcid.org/0000-0002-7699-2064

Reviewer Comments (if any, and for reference):

Reviewer's Responses to Questions

**Part I - Summary**

Reviewer #1: All of the reviewers comments have been addressed.

Reviewer #3: The authors outline the strengths and limitations of their study and of the ex vivo model used. Human tonsil and spleen organoids represent a relatively new and valuable tool for studying malaria immunology and vaccine immune responses, particularly in the context of γδ T-cell biology in human secondary lymphoid organs, a field hindered by limitations of animal models and immune-response analyses restricted to peripheral blood. By acknowledging the model limitations, addressing scientific and methodological concerns, and proposing future experiments during the revision process, the authors have adequately tackled the most relevant reviewers’ points. The manuscript is suitable for acceptance in its current revised form.

**Part II – Major Issues: Key Experiments Required for Acceptance**

Please use this section to detail the key new experiments or modifications of existing experiments that should be absolutely required to validate study conclusions.required to validate study conclusions.required to validate study conclusions.required to validate study conclusions.

Reviewer #1: (No Response)

Reviewer #3: The authors have adequately addressed the reviewers’ comments through clarification and additional discussion, without performing additional experiments.

**Part III – Minor Issues: Editorial and Data Presentation Modifications**

Reviewer #1: (No Response)

Reviewer #3: The authors have improved the manuscript’s clarity, editing, and data presentation.

PLOS authors have the option to publish the peer review history of their article (what does this mean?). If published, this will include your full peer review and any attached files.). If published, this will include your full peer review and any attached files.). If published, this will include your full peer review and any attached files.). If published, this will include your full peer review and any attached files.

...

Reviewer #1: No

Reviewer #3: No

---

## [Editor Report · Acceptance letter]

Dear Dr. Jagannathan,

We are delighted to inform you that your manuscript, "Cytotoxic Vδ2+ T cell subsets expand in response to malaria in human tonsil and spleen organoids," has been formally accepted for publication in PLOS Pathogens.

Best regards,

Sumita Bhaduri-McIntosh

Editor-in-Chief

PLOS Pathogens

orcid.org/0000-0003-2946-9497

Michael Malim

Editor-in-Chief

PLOS Pathogens

orcid.org/0000-0002-7699-2064